# ForecastPFN: Synthetically-Trained Zero-Shot Forecasting

**Samuel Dooley**[* 1]**, Gurnoor Singh Khurana**[1]**, Chirag Mohapatra**[1]**,
Siddartha Naidu**[1]**, Colin White**[1,2]

[1] Abacus.AI, [2] Caltech

## Abstract

The vast majority of time-series forecasting approaches require a substantial training dataset. However, many real-life forecasting applications have very little initial observations, sometimes just 40 or fewer. Thus, the applicability of most forecasting methods is restricted in data-sparse commercial applications. While there is recent work in the setting of very limited initial data (so-called 'zero-shot' forecasting), its performance is inconsistent depending on the data used for pretraining. In this work, we take a different approach and devise ForecastPFN, the first zero-shot forecasting model trained purely on a novel synthetic data distribution. ForecastPFN is a prior-data fitted network, trained to approximate Bayesian inference, which can make predictions on a new time series dataset in a single forward pass. Through extensive experiments, we show that zero-shot predictions made by ForecastPFN are more accurate and faster compared to state-of-the-art forecasting methods, even when the other methods are allowed to train on hundreds of additional in-distribution data points.

## 1 Introduction

Time-series forecasting is an important and long-studied problem that has attracted significant attention for many years [9, 14, 18, 29]. Time-series forecasting has a wide variety of applications such as healthcare [10, 20], economics [28, 42], climate science [5, 38], and renewable energy [19, 53]. Many forecasting algorithms have been proposed, including traditional statistical methods [4, 12], and more recently, deep learning methods [29, 30, 52, 54].

Nearly all time-series forecasting approaches (including both traditional methods and deep learning methods) require a substantial number of initial time-series datapoints in order to learn patterns and make future predictions. However, many real-world forecasting applications have very few initial observations, sometimes just 40 or fewer [6, 8, 16, 23]. This setting is deemed 'zero-shot forecasting' in prior work [36], in contrast to the typical setting in which hundreds or thousands of observations from the target series are used for training. While there is recent work on zero-shot forecasting [36], its performance is inconsistent depending on the dataset used for pretraining.

In this work, we take a different approach and devise ForecastPFN, the first **zero-shot** method trained purely on a **novel synthetic data distribution**. ForecastPFN is a prior-data fitted network (PFN) [33]; PFNs are a recently-proposed paradigm, in which a model is pretrained offline on synthetic data generated from a prior to approximate Bayesian inference. PFNs have recently been used in a breakthrough result for tabular datasets [22]. However, there are significant challenges when designing a general and flexible PFN in the forecasting setting, namely, constructing a general time-series distribution, and tuning an architecture and training scheme that can learn it.

---

[*]Correspondence to: `samuel@abacus.ai`, `crwhite@caltech.edu`.

37th Conference on Neural Information Processing Systems (NeurIPS 2023).

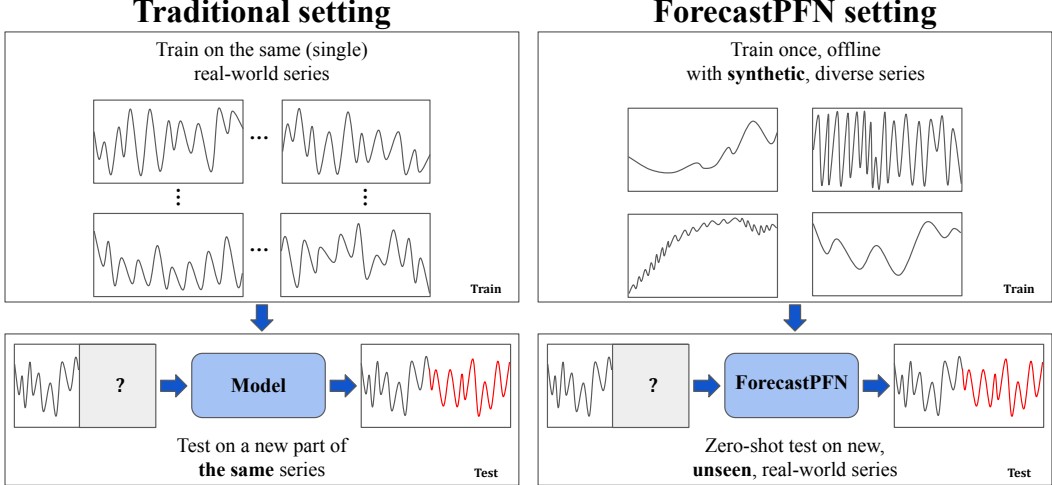

Figure 1: Left: standard forecasting setting, where a model trains on hundreds of datapoints for a time series, and tests on a future part of the same series. Right: ForecastPFN setting: a one-time, offline training routine on synthetic data, and then the model can make zero-shot predictions on new time series, without retraining or modifying the architecture weights.

We overcome these challenges first by designing a novel synthetic time-series distribution. We carefully design the synthetic distribution to be as general and diverse as possible, with a bias for common patterns seen in real-world series, while still being possible for a machine learning model to learn the distribution. Specifically, we design a modular synthetic model which incorporates multi-scale seasonal trends, linear and exponential global trends, and a Weibull-based noise distribution, each of which are parameterized with enough variance to capture a diverse range of time series, while still being possible to learn. Furthermore, using synthetic training data consisting of a ground-truth trend and a separate noise component means that we can remove the noise in the labels when calculating the train loss, which significantly improves the training speed of our model. Next, we design a flexible transformer architecture that can scale across a wide range of time-series values. While many existing transformers for forecasting train with a fixed prediction length and only predict the next $N$ timesteps into the future, ForecastPFN is trained to perform *arbitrary* queries – queries to any future timestep – allowing it to achieve zero-shot prediction for arbitrary prediction lengths.

We show that after a one-time, offline training phase on the synthetic dataset, ForecastPFN achieves strong *zero-shot* performance across a wide variety of real-world datasets, at a fraction of the runtime of other methods. ForecastPFN's zero-shot predictions outperforms state-of-the-art forecasting methods, *even when these other methods are allowed to train on hundreds of additional in-distribution datapoints*. For example, given a new time series, ForecastPFN makes future predictions after only seeing the input (typically the most-recent $\approx 30$ points of the series) and outperforms state-of-the-art methods which were allowed to train on an additional 300 historical points in the series (see Figure 1 and Figure 3). This remarkable result is due to the structure of our synthetic priors: by encoding multi-scale seasonal trends, global trends, and noise, across a variety of parameters, our model is able to learn how to forecast general time series. Finally, ForecastPFN is *fast*, producing predictions on a brand new dataset in just a single forward pass, taking 0.2 seconds. This is over 100 times faster than existing state-of-the-art transformer methods, which require expensive training procedures. Our codebase and our model are available at `https://github.com/abacusai/forecastpfn`.

**Our contributions.** We summarize our main contributions below.

- We introduce ForecastPFN, a prior-data fitted network for forecasting, trained purely on a novel synthetic data distribution. ForecastPFN is *zero-shot*: after its initial pretraining, it can make predictions on a brand new dataset with no training data from that dataset.
- Through extensive experiments across a variety of datasets, we demonstrate that ForecastPFN, without any retraining, outperforms state-of-the-art methods in low-resource settings, even when the other methods are allowed to train on additional in-distribution datapoints. Furthermore, ForecastPFN is fast, making predictions on a new dataset in a single forward pass.

## 2 Related Work

Time-series forecasting [9] has a multitude of applications throughout climate science [5, 38], healthcare [10, 20], business [7, 34], finance [27, 42], and renewable energy [19, 53]. A variety of approaches have been used for time-series forecasting [23, 29], including traditional statistical methods [4, 43] and deep learning methods [50, 52, 54].

ARIMA [4] is a statistical method from the 1970s that is still popular today, which builds an autoregressive model based on Markov processes. Another popular statistical method in practice is Prophet [45], which incorporates non-linear trends and multi-scale seasonality. Recently, deep learning methods for time series forecasting have become popular [29]. While early deep learning methods made use of RNNs [11, 41], transformer models have become popular more recently [26, 50, 52], following the success of transformers on NLP [47]. FEDformer [54] incorporates Fourier transforms and the seasonal trend decomposition method into a transformer architecture. For a survey on deep learning methods for time-series forecasting, see [29, 30], and for transfer learning for time-series forecasting, see [48].

Many real-world forecasting applications have very few initial observations, sometimes just forty or fewer [6, 8, 16, 23]. However, there is much less work in this setting (referred to as 'zero-shot' forecasting [36]). Oreshkin et al. [36] use a model based off of N-BEATS [35], a model that was state of the art at the time, training on a (single) real-world time series dataset and testing on a different time series. However, it achieves substantially different performance based on which training dataset is used. Another zero-shot forecasting method [37] uses an RNN base, yet it is only empirically compared to traditional methods and on a fixed prediction length. Recently, Adriaensen et al. [2] designed a zero-shot approach using a PFN for learning curve extrapolation; however, the setting for learning curve extrapolation is significantly different from time-series forecasting. To the best of our knowledge, we are the first to design a zero-shot time-series forecasting model trained purely on synthetic data. For additional related work, see Appendix A.

## 3 Prior-Data Fitted Networks for Forecasting

In this section, we give a background on prior-data fitted networks (PFNs), we introduce our synthetic data distribution, and we introduce ForecastPFN.

### 3.1 Background on Prior-Data Fitted Networks

A univariate time series is a sequence of observations collected over time, $\{(t, y_t)\}_{t \in \boldsymbol{t}}$, in which $t$ is a point in time, and $y_t$ is the value of the series at time $t$. In this section, for simplicity, we assume $\boldsymbol{t}$ is a set of integers $\{1, 2, \ldots, T\}$, so a time series can be written as $D = \{(t, y_t)\}_{t=1}^{T}$. Now let $\Phi$ denote a class of hypotheses, where each hypothesis $\varphi \in \Phi$ is a mechanism for generating a time series. For example, $\varphi$ may be defined by a function $\psi : A \to \mathbb{R}$, where $A \subseteq \mathbb{N}$, which generates an *underlying* base series; we generate the time series $D = \{(t, y_t)\}_{t=1}^{T}$ by $y_t = \psi(t) \cdot z_t$ for all $t \in A$, where $z_t$ is sampled from a noise distribution, $z_t \sim \mu$. In forecasting, we start with a partial time series $D = \{(t, y_t)\}_{t=1}^{T'}$, and the goal is to predict the value of the series, $\ell$ timesteps into the future.

The posterior predictive distribution (PPD) for a point $y$ is the distribution of its values given time $T$ and dataset $D = \{(t, y_t)\}_{t=1}^{T'}$ (where $T' < T$), denoted as $p(\cdot \mid T, D)$. It is computed for a particular point $y$ by integrating over the set of hypotheses $\Phi$ as follows:

$$p(y \mid T, D) \propto \int_{\Phi} p(y \mid T, \varphi) p(D \mid \varphi) p(\varphi) d\varphi. \tag{1}$$

A PFN $q_\theta$ is trained to approximate the PPD via *synthetic prior-fitting* [2, 33] as follows: iteratively sample a hypothesis $\varphi \sim p(\varphi)$ according to its prior probability, and then use $\varphi$ to sample a synthetic dataset $D \sim p(D \mid \varphi)$. We optimize the PFN's parameters $\theta$ by making predictions on a held-out output from $D$. We split the dataset into a training input set and a training output set. For the case of forecasting, that is $D_{\text{in}} = \{(t, y_t)\}_{t=1}^{\ell}$ and $D_{\text{out}} = \{(t, y_t)\}_{t=\ell}^{T}$. We use the PFN $q_\theta$ to predict $D_{\text{out}}$ given $D_{\text{in}}$. In the classification setting, we compute the cross entropy loss in expectation over the draw of the dataset $D \sim p(D)$, which trains the model to approximate the PPD in the long run [33]. Since forecasting is a regression task, we cannot use cross entropy loss, so we instead minimize

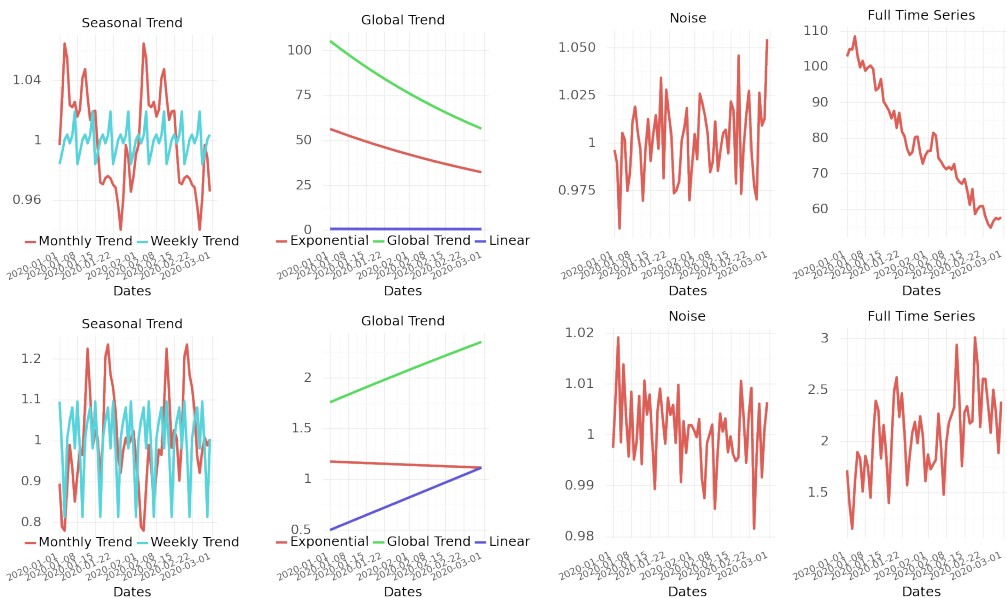

Figure 2: Two time series drawn from our synthetic distribution (Section 3.2). Each row shows the components that make up a time series: seasonal trends (far left), global trends (center left), and noise (center right), as well as the entire time series (far right). We plot only half of the series in order to show the seasonal trends clearly. See Appendix D for more examples.

the mean squared error loss $\mathcal{L}_\theta = \mathbb{E}_{D \sim p(D)} \left[ \sum_{t=\ell}^{T} (y_t - q_\theta(t, D_{\text{in}}))^2 \right]$, where $q_\theta(t, D_{\text{in}})$ denotes the model's point-prediction given $t$ and $D_{\text{in}}$. Over many samples draws from $p(\varphi)$, this procedure trains the model to predict the mean of the posterior distribution $p(\cdot \mid t, D_{\text{in}})$.

Note that the synthetic prior-fitting step is only done once, offline. At inference time, the resulting model is able to make *zero-shot* predictions: given a previously unseen input $D_{\text{in}}$, predict the corresponding $D_{\text{out}}$ (see Figure 1). The model's weights are not updated during inference: the model makes predictions for a new dataset in a single forward pass. We describe the details of our PFN architecture later in Section 3.3. In the next section, we describe our synthetic prior.

## 3.2 Defining a Synthetic Prior for Time Series

Unlike prior work in forecasting, our model is not trained on any real-world data; it is trained purely on synthetic data. As discussed above, we assume that there is a prior distribution of time series, $p(\varphi)$, from which real-world time series are derived. Intuitively, this distribution includes components such as periodic (weekly, monthly, and yearly) trends, global trends, and noise. We show that when we model data from the distribution described below, we can adequately capture aspects of time-series data in order to produce a high-performance, zero-shot forecasting model.

We model our synthetic data with the simple premise that time series data have two independent components: underlying ($\psi$) and noise ($z_t$). We include time series for three scales: daily, monthly, and yearly, i.e., series where an observation is taken once a day, once a month, or once a year. Data are generated from each periodicity independently according to the following procedure. Now we describe daily series, and we describe the monthly and yearly time series in Appendix D.

We model the underlying time series as being comprised of a seasonal and a trend component. We see these as three independent aspects of time series data and model them as below, where the time series is the product of a trend and seasonality with an additional noise factor. The trend component is made up of a linear and exponential component with coefficients $m_{\text{lin}}, c_{\text{lin}}, m_{\text{exp}}, c_{\text{exp}}$. The seasonal component has a week, month, and year component where each comprises of coefficients $p_{\text{week}}, p_{\text{month}}, p_{\text{year}}$ Finally, the noise in our model is derived from a Weibull distribution and is modeled such that the expected value of the noise model is 1, meaning in expectation the noise does not contribute to the seasonality or trend of the time series. Formally,

$$y_t = \psi(t) \cdot z_t = \text{trend}(t) \cdot \text{seasonal}(t) \cdot z_t, \text{ where}$$

$$z_t = 1 + m_{\text{noise}}\left(z - \bar{z}\right), \text{ where } z \sim \text{Weibull}(1, k), \ \bar{z} = (\ln 2)^{1/k}$$

$$\text{trend}(t) = (1 + m_{\text{lin}} \cdot t + c_{\text{lin}})\left(m_{\text{exp}} \cdot c_{\text{exp}}^t\right)$$

$$\text{seasonal}(t) = \text{seasonal}_{\text{week}}(t) \cdot \text{seasonal}_{\text{month}}(t) \cdot \text{seasonal}_{\text{year}}(t), \text{ where}$$

$$\text{seasonal}_\nu(t) = 1 + m_\nu \sum_{f=1}^{\lfloor p_\nu/2 \rfloor}\left[c_{f,\nu}\sin\left(2\pi f \frac{t}{p_\nu}\right) + d_{f,\nu}\cos\left(2\pi f \frac{t}{p_\nu}\right)\right],$$

$$\text{where } \nu \in \{\text{week}, \text{month}, \text{year}\}, \quad p_{\text{week}} = 7, \ p_{\text{month}} = 30.5, \ p_{\text{year}} = 365.25.$$

We choose multiplicative noise to better balance the amount of signal to noise across all series. If we had used additive noise, then the impact of the noise would have depended on the ratio of the scale of the base series and the scale of the noise. Since our base series have linear and exponential terms, additive noise would cause the signal-to-noise ratio to vary substantially over series based on the trend component. Furthermore, the Weibull distribution is a simple and natural method to parameterize between Gaussian and exponential distributions, two types of distributions that frequently come up in real-world time series.

The above synthetic data generation model is extremely general and can capture a variety of time series we would encounter in real world forecasting applications. The model contains a number of parameters, which we denote as

$$\boldsymbol{\xi} = \{m_{\text{lin}}, c_{\text{lin}}, m_{\text{exp}}, c_{\text{exp}}, m_{\text{noise}}, c_{1,\text{week}}, d_{1,\text{week}}, \ldots, c_{p_{\text{year}}2/,\text{year}}, d_{p_{\text{year}}/2,\text{year}}\}.$$

Our full prior $p(\varphi)$ consists of a product distribution over all elements of $\boldsymbol{\xi}$, where

$$m_{\text{lin}}, m_{\text{exp}} \sim \mathcal{N}(\mu_m, \sigma_m); \quad c_{\text{lin}} \sim \mathcal{N}(0, \sigma_{\text{lin}}); \quad c_{\text{exp}} \sim \mathcal{N}(1, \sigma_{\text{exp}}).$$

For all pairs of $\nu$ and $f$, $c_{f,\nu}$ and $d_{f,\nu}$ are both drawn from $\mathcal{N}(0, 1/f)$, such that these coefficients are inversely proportional to the harmonics of the series, and then they are evenly rescaled so that the sum of their squares equals 1.

Therefore, our synthetic data generation model contains six hyperparameters: $\mu_m, \sigma_m, \sigma_{\text{lin}}, \sigma_{\text{exp}}$ define the trend component, and $m_{\text{noise}}, k$ define the noise component. See Figure 2 for examples of daily time series drawn from $p(\varphi)$, broken down into their different components.

### 3.3 ForecastPFN: a PFN for Zero-Shot Forecasting

Now we give the details for designing ForecastPFN, the first zero-shot forecasting method trained purely on synthetic data. We focus on the novel components inspired by the extreme challenge of training across a wide diversity of time-series scales with a single model.

**Architecture Details.**  As with the original PFNs [2, 22, 33], we use a transformer [47] as the base architecture. We use an encoder-based transformer, consisting of one multi-head attention layer and two feedforward layers. This is in contrast to prior work on zero-shot forecasting, which used residual networks [36] or recurrent neural networks [37].

The transformer accepts data in the form of tokens $(t, y_t)$. The time stamps $t$ as part of the input data tokens are represented in terms of temporal features such as the year, month, day, day of the week, and day of the year. The series values are scaled as described in the robust scaler below. A query for ForecastPFN is extremely general: the transformer takes in a set of tokens $(t, y_t)$, along with a query consisting of a **single, arbitrary** future date, and then it predicts the output of this query. Thus, the input to the model is a set of tokens and the output is a single prediction at a future time, specified by the user. The input does not need to be a fixed size, and the input tokens do not need to be contiguous.

The generality of ForecastPFN queries allows it to perform well at test time, on a very diverse range of datasets and settings. This is also in stark contrast to existing transformer models for forecasting, which typically are only set up to predict the next $N$ steps in the current sequence.

**Robust Scaling.** One of the most challenging technical problems when training ForecastPFN is handling the extreme range of the scales of the time series, both in terms of the absolute values and the trends. This is a challenge both at training time and when computing the prediction of a time series, and partially stems from the inclusion of the multiplicative, exponential, and noise terms in our synthetic generation model. Conventional scaling techniques such as standardization (Z-score normalization) and min-max scaling techniques cannot apply: for example, not only does the absolute range differ greatly across training samples, but there is also large diversity in the number of outliers, and the scale of the pattern (i.e., periodicity and global trends) relative to the absolute range.

To handle these challenges, we perform three steps of outlier removal: *(1)* mask out any missing values, *(2)* standardize the data based on all non-outlier datapoints (defined as 2-sigma outliers), and *(3)* clip all 3-sigma outliers. This procedure allows us to put all time series into a relatively consistent range, while removing outliers that cause exploding gradients. Furthermore, when computing the loss of a time series, we scale the mean squared error (MSE) based on the maximum of the (scaled) input. The effect is a loss function that is in between MSE, which gives too much weight for series with a high range, and mean squared percentage error (MSPE), which gives too much weight for series with a small range. We show in Section 4 that robust scaling significantly boosts performance.

**Training Details.** We train ForecastPFN using the synthetic prior $p(\varphi)$ defined in Section 3.2. During early-stage model development, we chose hyperparameters $\mu_m, \sigma_m, \sigma_{\text{lin}}, \sigma_{\text{exp}}, m_{\text{noise}}, k$ to create the most diverse data distribution possible, with the model still being able to train without diverging or stalling. For example, when the parameters defining the noise distribution become too large, the training data is too noisy and the model cannot converge. We specifically note that *(1)* all model development was done *before* evaluating on the seven real-world datasets defined the next section, and *(2)* all experiments were done with a single ForecastPFN model (specifically, after developing the model, we trained it once, and then used it for all experiments in the next section.

Crucially, since we use synthetic training data with a ground-truth trend and a separate noise component, we can remove the noise when calculating the train loss. This significantly improves the training speed of the model, since the model cannot learn from the noise, as it is independent of time (Appendix E.5).

We set the input length $\ell = 100$ and the maximum prediction of 10 steps into the future (note: in the next section, we see that ForecastPFN is able to make accurate predictions significantly beyond these settings). We generate 300 000 synthetic series to use as training data, each of length 200, and we use a sliding window of size 100 to obtain 101 prediction tasks per generated series. Each epoch consists of 1 024 000 tasks, and we trained the transformer for 600 epochs with the Adam optimizer [25] on a single Tesla V100 16GB GPU, which took 30 hours. This training is done offline, and only once, and we open-source the trained model at `https://github.com/abacusai/forecastpfn`. For the full development, implementation, and training details, see Appendix E.

## 4 Experiments

We demonstrate the power of ForecastPFN by comparing it to various high-performing forecasting models across several datasets. We compare all models across different training time budgets, and we compare the zero-shot performance of ForecastPFN to other models with different data budgets.

**Algorithmic Baselines.** We compare against three simple numerical baselines: Last, Mean, and SeasonalNaive [23]; two high-performing traditional methods: ARIMA [4] and Prophet [23]; three state-of-the-art transformer-based methods: FEDformer [54], Autoformer [50], and Informer [52]; a vanilla Transformer method [47]; and one state-of-the-art zero-shot method: Meta-N-BEATS [36]. All of the methods were implemented using their default hyperparameters and official codebase, with the exception of ARIMA (which has no 'official' codebase), for which we use `pmdarima` [43].

**Benchmark Datasets.** To ensure a fair comparison, we evaluate on seven popular, real-world datasets across energy systems, economics, traffic, and weather: ECL (Electricity Consuming Load) [46], ETT 1 and 2 (Electricity Transformer Temperature) [52], Exchange [28], Illness [17], Traffic [39], and Weather [1]. These datasets are standard for benchmarking the performance of forecasting methods and are the same ones used by many of our baselines [36, 50, 52, 54]. The datasets are described in the Appendix E.

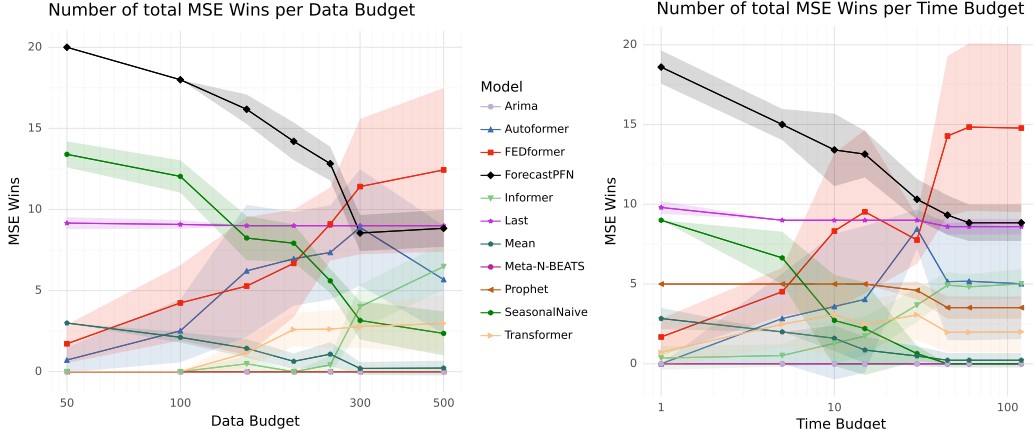

Figure 3: Analysis of performance vs. **data budget** and **time budget**, aggregated across datasets and prediction lengths. We plot the number of total MSE wins (left) where a higher value is better. Error bars show one standard deviation across training runs. Recall, ForecastPFN and Meta-N-BEATS see no training data for these series, only the length 36 input.

**Experimental Setup.** The two zero-shot methods, ForecastPFN and Meta-N-BEATS, were both trained offline before the evaluations on benchmark datasets. We train ForecastPFN **once** on our synthetic data, as described in Section 3, and then that model is used for all evaluations in this paper. For Meta-N-Beats, we use the best version as described in their paper [36], which is the one trained on M4 [31]. However, since the prediction length needs to be specified upfront, we train several Meta-N-Beats models, one for each prediction length considered below. For each time series, the zero-shot models see only the input of length 36 and then predict future timesteps. These training regimes are different from the other, non-zero shot models which are trained for each configuration.

We compare the zero-shot methods to the nine other non-zero-shot methods described above, even though this is not a fair comparison: the non-zero-shot methods are allowed to train on earlier portions of the time series used at test time, as is common in standard forecasting algorithms. We explore the performance of these algorithms across various resource constraints in two distinct directions: by restricting the training **time budget** and by restricting the amount of training **data budget** given to the non-zero-shot methods. We restrict the data budget of the non-zero-shot methods to lengths from 50 to 500. We restrict the time budget of the non-zero-shot methods to wall clock budgets from 1 to 120 seconds, where timing is calculated after data loading and only during the training loop.

These two restrictions follow naturally from the ways in which forecasting datasets and applications can be constrained in the real world. Many real-world forecasting applications have very few initial observations [6, 8, 16, 23]. For example, many businesses need demand forecasts or customer churn forecasts for a large number of products or customers, respectively, which are only size 100 daily series or size 36 monthly series. Furthermore, there is also a wide range of applications which require a low time (or computational) budget in resource constrained environments, such as on a CPU or on edge devices. For example, forecasting may be needed to predict energy demand in developing countries. Another application is as a forecasting data-exploration tool, allowing users to see instant forecasting visualizations as they navigate their dataset.

We also test all algorithms with different prediction lengths, or the number of steps into the future an algorithm must predict. Most deep-learning-based algorithms require this hyperparameter to be set before training, and then that prediction length is encoded into the algorithm's architecture. This is true for FEDFormer, Informer, Autoformer, the vanilla transformer, and Meta-N-BEATS, so we retrain these models for each experiment with a new prediction length. We choose prediction lengths from 6 to 48 as in prior work [54]. At test time, all algorithms are given series with input length of 36 and asked to predict into the future at these variable prediction lengths. For example, say that we have a time series $\{(t, y_t)\}_{t=1}^{600}$, a data budget of $x$, and a prediction length of $\ell$. We allow algorithms to use 10% of their data budget on validation. All non-zero-shot methods are allowed to train on $\{(t, y_t)\}_{t=500-x}^{500}$. Then, at test time, *all* algorithms see the 36 input data points and make a prediction length of $\ell$, e.g., input of $\{(t, y_t)\}_{t=501}^{536}$ and make predictions for timesteps $t = 537$ to $537 + \ell$.

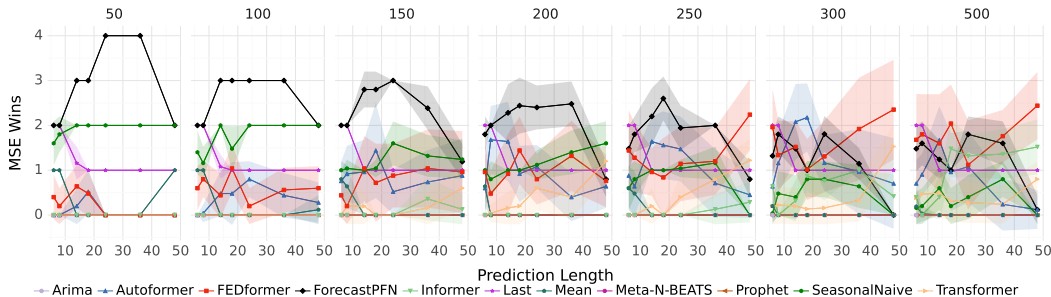

Figure 4: Analysis of performance across prediction lengths for increasing data budgets (left subplot, data budget is 50 to the right subplot, data budget is 500), aggregated across datasets. Total MSE wins for each scenario is plotted with error bars as one standard deviation across training runs. Recall, ForecastPFN and Meta-N-BEATS see no training data for these series, only the length 36 input.

|  | | ECL | ETTh1 | ETTh2 | Exchange | Illness | Traffic | Weather |
|---|---|---|---|---|---|---|---|---|
| **Data Budget = 50** | Arima | 1.840 | 0.341 | 1.610 | 1.175 | 5.045 | 2.738 | 0.042 |
| | Autoformer | 1.289 | 0.570 | 1.107 | 0.331 | 1.353 | 2.119 | 0.520 |
| | FEDformer | 0.683 | 0.403 | 1.020 | 0.233 | 1.240 | 1.318 | 0.330 |
| | ForecastPFN | 1.075 | **0.127** | **0.330** | 0.058 | 1.091 | 1.971 | **0.009** |
| | Informer | 1.252 | 0.982 | 0.890 | 2.834 | 8.584 | 3.627 | 0.436 |
| | Last | 0.910 | 0.176 | 0.466 | **0.022** | **1.077** | 3.081 | 0.014 |
| | Mean | 0.673 | 0.158 | 0.600 | 0.040 | 1.091 | 1.585 | 0.011 |
| | Meta-N-BEATS | 0.909 | 0.177 | 0.480 | 0.023 | 1.301 | 2.913 | 0.014 |
| | Prophet | 2.174 | 3.298 | 14.135 | 421.635 | 11.824 | 2.352 | 0.101 |
| | SeasonalNaive | **0.453** | 0.203 | 0.554 | 0.028 | 1.410 | **0.653** | 0.017 |
| | Transformer | 0.945 | 0.516 | 1.043 | 0.941 | 4.320 | 2.244 | 0.244 |
| **Data Budget = 500** | Arima | 1.969 | 0.200 | 1.099 | 1.175 | 4.848 | 1.661 | 0.042 |
| | Autoformer | 0.513 | 0.144 | 0.338 | 0.072 | 0.737 | 0.526 | 0.210 |
| | FEDformer | 0.480 | 0.133 | 0.352 | 0.068 | **0.707** | **0.523** | 0.188 |
| | ForecastPFN | 1.075 | **0.127** | 0.330 | 0.058 | 1.091 | 1.971 | **0.009** |
| | Informer | **0.453** | 0.144 | **0.253** | 0.529 | 4.394 | 1.084 | 0.224 |
| | Last | 0.910 | 0.176 | 0.466 | **0.022** | 1.077 | 3.081 | 0.014 |
| | Mean | 0.673 | 0.158 | 0.600 | 0.040 | 1.091 | 1.585 | 0.011 |
| | Meta-N-BEATS | 0.909 | 0.177 | 0.480 | 0.023 | 1.301 | 2.913 | 0.014 |
| | Prophet | 15.668 | 3.228 | 4.960 | 13.346 | 3.462 | 1.556 | 0.057 |
| | SeasonalNaive | **0.453** | 0.203 | 0.554 | 0.028 | 1.410 | 0.653 | 0.017 |
| | Transformer | 0.541 | 0.139 | 0.274 | 0.280 | 3.482 | 0.935 | 0.016 |

Table 1: Average MSE values for each algorithm and each dataset on the shortest (50) and longest (500) data budgets. We observe that ForecastPFN is the majority winner on the smaller data budget and tied for best on the longer data budget. Thus, we see the robustness of the ForecastPFN method across a range of datasets and data budgets.

We conduct training runs under the Cartesian product of each prediction length and each data budget, as well as the Cartesian product of each prediction length and each time budget. We repeat these runs on every dataset and every algorithm. We conduct each run five times with different random seeds (in the randomness of the algorithm). However, ForecastPFN, Meta-N-BEATS, ARIMA, Prophet, and the baselines are deterministic algorithms, so they are only evaluated with one seed per run.

We plot the MSE in this section, and we give additional metrics in Appendix E. In order to aggregate results among all datasets and different random seeds, for each experiment, we compute the number of 'wins' (rank of 1) for each algorithm, as is commonly done in prior work [22, 33]. We plot the performance across data and time budgets in Figure 3, aggregated over datasets and prediction lengths, and tabulate the MSE of each model on each dataset in Table 1. Additionally, we plot the change in

performance for the prediction lengths on each data budget in Figure 4 and in Appendix E). We plot error bars as one standard deviation from the mean by averaging over the training runs.

**Results and Discussion.** We find that ForecastPFN significantly outperforms all ten other methods in the low data budget and low time budget settings (Figure 3). Specifically, we see that ForecastPFN is the best architecture by the number of MSE wins for time budgets 1 to 30. Similarly, ForecastPFN is the best architecture by the number of MSE wins for data budgets from 50 to 250. In the settings with higher data or time budget, ForecastPFN is still competitive with state-of-the-art methods. This story continues to hold across prediction lengths as well in Figure 4. ForecastPFN is the outright winner in nearly every prediction length for data budgets 50, 100, and 150, and becomes either outright best or very competitive in nearly every prediction length for budgets 200, 250, and 300. Additionally, we see from Table 1 that ForecastPFN achieves the lowest average MSE value on more datasets than any other algorithm, at a data budget of 50, and remains competitive at a data budget of 500. We also find that the simple baselines, SeasonalNaive and Last, often achieve second place behind ForecastPFN in the lowest training and runtime settings. This is not unexpected and is a testament to the challenge of predicting a series when given very little training data or training time.

The remarkable performance of ForecastPFN is despite the fact that, as explained above, the zero shot methods are at a disadvantage because the other methods were allowed to train on data from earlier in the time series. When ForecastPFN is compared to Meta-N-Beats, the only fair comparison, we see that ForecastPFN strictly outperforms Meta-N-Beats across all settings. We believe these strong results are due to our synthetic prior: by encoding multi-scale seasonal trends, global trends, and noise, across a variety of parameters, ForecastPFN learns very strong prior knowledge about the common patterns in time series. While $\approx 250$ datapoints of a series contain enough complexity and noise for the training-based algorithms to struggle to make accurate predictions when learning the series 'from scratch', ForecastPFN's prior knowledge from synthetic series is significant enough to make better predictions when only given access to the 36 input points.

Furthermore, ForecastPFN is significantly faster than the other transformer-based methods such as FEDformer, Informer, and Autoformer. ForecastPFN makes predictions in a single forward pass, just 0.2 seconds; in order to reach the same accuracy as ForecastPFN, the other transformer-based methods need take over 100 times more runtime.

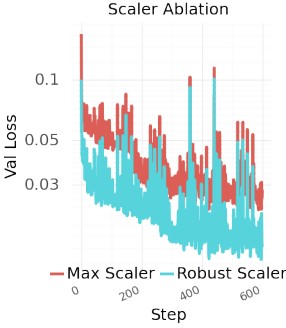

Figure 5: Robust scaling vs. min-max scaling.

| | Epoch | 0 | 100 | 200 | 300 |
|---|---|---|---|---|---|
| Train | ForecastPFN, lowest noise | 23.71 | 1.484 | 0.878 | 0.591 |
| | ForecastPFN, low noise | 0.359 | 0.050 | 0.039 | 0.384 |
| | ForecastPFN | **0.207** | **0.049** | **0.037** | **0.032** |

Table 2: Loss values reported for three training runs of ForecastPFN where the amount of noise used to synthetically generate the ForecastPFN training data is moderated. In this table, we scale $m_{\mathrm{noise}}$ to $1/6$ for the lowest noise model, to $2/3$ for the low noise model, and 1 for the ForecastPFN model.

**Ablation Studies.** Since ForecastPFN takes 30 GPU hours to train, we do not conduct substantial ablation studies. However, we are able to investigate two important aspects of the ForecastPFN algorithm: our custom scaler and our noise model.

First, we investigate the impact of robust scaling (described in Section 3.3) on the performance of ForecastPFN. We compare robust scaling to min-max scaling. We conduct a training run of ForecastPFN with each scaler and plot the validation MSE loss on the synthetic dataset in Figure 5. We can clearly see that the robust scaler achieves a lower validation loss throughout training. We conclude the use of our robust scaler mitigates the effect of extreme values. While the presence of extreme values make it harder for the model to differentiate small-scale trends in a series, our robust scaler allows for better, faster convergence.

Finally, we investigate the impact of noise in our synthetic training data generation process (described in Section 3.3). We trained ForecastPFN with three scales of our noise parameter, $m_{\mathrm{noise}}$: 1 (original

ForecastPFN), 2/3 (low noise), and 1/6 (lowest noise). We do not change the scale of the noise in the validation data. In Table 2, we report the train and validation MSE on synthetic data for each model. Recall that we remove noise in the ground-truth predictions of the training data as a design decision that improves performance, which is why the scale looks different for the train and validation losses. We also look at removing the noise entirely in Appendix E.5 and come to similar conclusions about the level of noise. We can conclude from this ablation that the noise parameters we chose in our experiments provide for robust minimal train and validation losses, validating our approach.

## 5 Conclusions, Limitations, and Future Work

In this work, we introduced ForecastPFN, the first zero-shot forecasting model that is trained purely on synthetic data. Our novel synthetic data distribution incorporates multi-scale seasonal trends, linear and exponential global trends, and a noise distribution, capturing an extremely diverse range of time series. ForecastPFN is a prior-data fitted network, trained once, offline to approximate Bayesian inference, which can make predictions on a new time series dataset in a single forward pass. Through extensive experiments, we show that zero-shot predictions made by ForecastPFN are more accurate and faster compared to state-of-the-art forecasting methods, even when the other methods are allowed to train on hundreds of additional in-distribution data points.

**Limitations and Future Work.** While ForecastPFN has shown remarkable success in the zero-shot setting, there are still limitations as well as exciting directions left for future work. Due to the vanilla transformer architecture, ForecastPFN is limited by its training data to time series of fewer than 1000 points. However, given the recent advances in scaling transformers [3, 13, 26, 44], this is not inherently a problem for ForecastPFN. Currently, ForecastPFN only works for univariate time series. Once again, this is not an inherent limitation, and designing ForecastPFN for multivariate time series is a promising direction for future work. Furthermore, using exogenous features are interesting ideas for follow-up work.

Our synthetic prior was created to focus on 'human-like' or 'earth-like' time series (daily, weekly, yearly periods), so ForecastPFN may not not perform well on uncommon periods such as 41, 89, or 97. ForecastPFN also only approximates the posterior predictive distribution and makes pointwise predictions. Outputting probabilistic predictions are an exciting area for future work. Finally, ForecastPFN is particularly well-suited to handling missing data (including large gaps), or varying time frequencies within the same series, in contrast to existing forecasting methods. An exciting future direction is to explore the potential to forecast time series in which the input contains large gaps or varying frequencies.

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

# A  Additional Related Work

Time-series forecasting [9] has a multitude of applications such as healthcare [10, 20], climate science [5, 38], business [7, 34], finance [27, 42], and renewable energy [19, 53]. A variety of approaches have been used for time-series forecasting [23, 29], including traditional statistical methods [4, 43] and deep learning methods [50, 52, 54].

**Forecasting.**  ARIMA [4] is a statistical method from the 1970s that is still popular today, which builds an autoregressive model based on Markov processes. Another popular statistical method in practice is Prophet [45], which incorporates non-linear trends and multi-scale seasonality. Recently, deep learning methods for time series forecasting have become popular [29]. While early deep learning methods made use of RNNs [11, 41], transformer models have become popular more recently [26, 50, 52], following the success of transformers on NLP [47]. FEDformer [54] incorporates Fourier transforms and the seasonal trend decomposition method into a transformer architecture. For a survey on deep learning methods for time-series forecasting, see [29, 30].

Many real-world forecasting applications have very few initial observations, sometimes just forty or fewer [6, 8, 16, 23]. However, there is much less work in this setting (referred to as 'zero-shot' forecasting [36]). Oreshkin et al. [36] use a model based off of N-BEATS [35], a model that was state of the art at the time, training on a (single) real-world time series dataset and testing on a different time series. However, it achieves substantially different performance based on which training dataset is used. Another zero-shot forecasting method [37] uses an RNN base, yet it is only empirically compared to traditional methods and on a fixed prediction length.

There is some work on transfer learning for time-series forecasting, but it is still limited [24]. Earlier works study transfer learning by pretraining a CNN architecture for forecasting on one time series, and fine-tuning on another time series [15, 51]. Fawaz et al. showed that some datasets can improve performance, while others can degrade performance, and they introduce a new method to predict the best source dataset. More recent work uses LSTMs and CNN+RNN hybrid models, to achieve better results [48]. Global Forecasting Models (GFM) are a new type of forecasting model in which a single model is trained on all series from a single dataset [21, 40, 41, 49]. However, a key assumption is that these time series must be related in some way. For example, a retail company may train a single model for the time series of sales of thousands of different products.

**Prior-Data Fitted Networks.**  Prior-data fitted networks (PFNs) are a recently-proposed paradigm for machine learning, in which a model is trained offline on small, synthetic datasets. Once trained, a PFN can infer the posterior predictive distribution (PPD) for test datapoints, given a new dataset, in a single forward pass.

Müller et al. [33] first introduced PFNs, showing that they provably approximate Bayesian inference, and demonstrating performance on very small datasets. Soon after, TabPFN [22] showed state-of-the-art empirical performance on datasets of up to size 1000 [32]. This new empirical finding was accomplished in part due to the synthetic prior, consisting of a large set of structural causal models. Recently, Adriaensen et al. [2] designed a zero-shot approach using a PFN for learning curve extrapolation; however, the setting for learning curve extrapolation is significantly different and arguably a special case of time-series forecasting. To the best of our knowledge, we are the first to design a zero-shot time-series forecasting model trained purely on synthetic data.

# B  Broader Societal Impact

We do not see any strongly negative broader societal impacts of this work. In fact, our hope is that this work will reduce the computational cost and $CO_2$ emissions for forecasting. Although ForecastPFN takes 30 GPU hours to train, this is a one-time cost, which we have already done. Practitioners can now download and use our model, rather than training methods such as FedFormer and Informer from scratch. In general, we hope that our work encourages practitioners and researchers to conduct zero-shot (or few-shot) forecasting, rather than training models from scratch for each new dataset.

## C   Reproducibility

Our codebase is available at https://github.com/abacusai/forecastpfn. The README to the codebase contains instructions for installation and immediately using ForecastPFN to make predictions on a new dataset. We also include all code from this project, including the code for synthetic data generation, ForecastPFN training, and evaluation.

## D   Synthetic Data Details

We give the full details of the synthetic data series, including the parameters for daily, weekly, and monthly series.

Recall that the time series has a trend component, seasonal component, and noise component. The trend component is made up of a linear and exponential component with coefficients $m_{\text{lin}}, c_{\text{lin}}, m_{\text{exp}}, c_{\text{exp}}$. The seasonal component has a week, month, and year component where each comprises of of coefficients $p_{\text{week}}, p_{\text{month}}, p_{\text{year}}$ Finally, the noise in our model is derived from a Weibull distribution and is modeled such that the expected value of the noise model is 1, meaning in expectation the noise does not contribute to the seasonality or trend of the time series. Formally,

$$y_t = \psi(t) \cdot z_t = \text{trend}(t) \cdot \text{seasonal}(t) \cdot z_t, \text{ where}$$

$$z_t = 1 + m_{\text{noise}}\left(z - \bar{z}\right), \text{ where } z \sim \text{Weibull}(1, k), \ \bar{z} = (\ln 2)^{1/k}$$

$$\text{trend}(t) = (1 + m_{\text{lin}} \cdot t + c_{\text{lin}})\left(m_{\text{exp}} \cdot c_{\text{exp}}^t\right)$$

$$\text{seasonal}(t) = \text{seasonal}_{\text{week}}(t) \cdot \text{seasonal}_{\text{month}}(t) \cdot \text{seasonal}_{\text{year}}(t), \text{ where}$$

$$\text{seasonal}_\nu(t) = 1 + m_\nu \sum_{f=1}^{\lfloor p_\nu/2 \rfloor} \left[c_{f,\nu} \sin\left(2\pi f \frac{t}{p_\nu}\right) + d_{f,\nu} \cos\left(2\pi f \frac{t}{p_\nu}\right)\right].$$

Now we explain how to set all unspecified parameters above, for daily, weekly, and monthly series. For convenience, when creating a daily, weekly, or monthly series, a unit of time $t$ is equal to 1, 7, or 30.5 days (which would be equivalent to scaling all the parameters below by 1, 7, or 30.5).

For all pairs of $\nu$ and $f$, $c_{f,\nu}$ and $d_{f,\nu}$ are both drawn from $\mathcal{N}(0, 1/f)$, such that these coefficients are inversely proportional to the harmonics of the series, and then they are evenly rescaled so that the sum of their squares equals 1.

We set each $m_\nu$ for $\nu \in \{\text{week}, \text{month}, \text{year}\}$ separately for daily, weekly, and monthly, as follows. $m_\nu = 0$ unless specified:

- Daily: $m_{\text{week}} \sim U([0, 1])$, $m_{\text{month}} \sim U([0, .2])$, $p_{\text{week}} = 7$, $p_{\text{month}} = 30.5$.

- Weekly: $m_{\text{month}} \sim [0, .3]$, $m_{\text{year}} \sim U([0, .1])$, $p_{\text{month}} = 2$, $p_{\text{year}} = 52$.

- Monthly: $m_{\text{year}} \sim U([0, .5])$, $p_{\text{year}} = 12$.

Finally,

$$m_{\text{lin}}, m_{\text{exp}} \sim \mathcal{N}(-.01, 0.5); \quad c_{\text{lin}} \sim \mathcal{N}(0, .01); \quad c_{\text{exp}} \sim \mathcal{N}(1, 0.005).$$

As described in Section 3.3 and the next section (Appendix E.1), we set these parameters to create the most diverse data distribution possible, with the model still being able to train without diverging or stalling. For example, when the parameter for the noise distribution becomes too large, the model cannot converge. We also released a README for our synthetic data generation at https://github.com/abacusai/forecastpfn.

For visualizations of daily time series drawn from this distribution, see Figure 2 and Figure 7.

# E  Additional Experimental Details

## E.1  ForecastPFN Development

We estimate that the total computational budget used in this project, from initial research to running all the final evaluations, is 500 GPU hours. Our computation consisted mainly of designing the architecture and designing the synthetic data distribution. In this design phase, we used the train loss of the synthetic data as a signal, because many synthetic data distributions, and architectures, cause the model not to train, for example, setting the noise parameter too high. We also used the synthetic held-out validation set to make decisions, such as in Figure 5 and Appendix E.5.

The training of our ForecastPFN model is 30 GPU hours on a single Tesla V100 16GB GPU. We emphasize that this model training is only done once, and we released the trained ForecastPFN model.

For an ablation study, we tried to achieve non-trivial zero-shot performance with the FedFormer architecture. However, after significant effort, we were unable to achieve non-degenerate performance with the FedFormer architecture.

## E.2  Implementation Details

**ForecastPFN Implementation Details.**  We use an encoder-based transformer architecture, with two encoders. Each encoder consists of a multi-head attention layer with four heads, and two feedforward layers. The first feedforward layer is size 32 times the embedding dimension, and the second feedforward layer is size 8 times the embedding dimension.

**Baseline Implementation Details.**  We compare ForecastPFN against ARIMA [4], Autoformer [50], FEDformer [54], Informer [52], Meta-N-BEATS [36], Prophet [45], and a vanilla Transformer method [47]. All of the methods were implemented using their default hyperparameters and official codebase. For ARIMA (which was designed in the 1970s and has no 'official' codebase), we used `pmdarima` [43]. The full implementation details of all methods can be found in our open source codebase. Furthermore, we compared to three baselines: Last (prediction is the last seen value in the series), Mean (prediction is the mean value in the series), and SeasonalNaive (prediction is the mean value from all prior instances of the same day of the week).

In Section 4, in the experiments with a restricted time budget, we restrict the time budget of the non-zero-shot methods to wall clock budgets from 1 to 120 seconds, where timing is calculated after data loading and only during the training loop. After each training step, we stop the training if the total budget was exceeded. In case an algorithm is unable to output any predictions within the data budget (such as ARIMA with a 1 second budget), we set the output to 0's.

## E.3  Training Details

We generate 100 000 each of synthetic daily, weekly, and monthly series, each of length 200, and we use a sliding window of size 100 to obtain 101 prediction tasks per series. We sample 1024 tasks in a training step, and there are 1000 training steps in an epoch. We trained the transformer for 300 epochs on a V100 16GB GPU, which took 30 hours.

We use the Adam optimizer with a learning rate of 0.0001, and MSE loss. We set the maximum prediction into the future to be 10 times the frequency of the training series.

## E.4  Dataset Details

We use seven datasets for our experiments in Section 4. These datasets are standard for benchmarking the performance of forecasting methods and are the same ones used by the majority of our baselines [36, 50, 52, 54].

Illness is a dataset [17] containing influenza-like illness patients in the United States. It contains 966 datapoints, spaced weekly. Exchange is a currency exchange dataset [28], consisting of 7588 datapoints, spaced daily. Traffic is a dataset consisting of freeway congestion across California [39]. ETT (Electricity Transformer Temperature) is a electricity power deployment dataset [52] across 2 years from a power plant in China. The target value is oil temperature. Two separate series

| Time Budget | 1.0 | 5.0 | 10.0 | 15.0 | 30.0 | 45.0 | 60.0 | 120.0 |
|---|---|---|---|---|---|---|---|---|
| Arima | 9.68 | 77.42 | 88.71 | 92.9 | 100 | 100 | 100 | 100 |
| Autoformer | 70.97 | 97.74 | 100 | 100 | 100 | 100 | 100 | 100 |
| FEDformer | 71.94 | 99.68 | 100 | 100 | 100 | 99.35 | 100 | 100 |
| Informer | 72.58 | 100 | 100 | 100 | 100 | 100 | 97.42 | 100 |
| Prophet | 88.71 | 100 | 100 | 100 | 100 | 100 | 100 | 100 |
| Transformer | 66.77 | 100 | 100 | 100 | 98.39 | 100 | 100 | 100 |

Table 3: The percentage of configurations (dataset, prediction lengths, seeds) that failed, for each time budget and each model which requires training or being fit.

are provided: ETT1 and ETT2. Each series represents data collected from two different electicity transformer stations. ECL (Electricity Consuming Load) [46] is an electricity consumption (Kwh) dataset across two years with a resolution of every hour. This dataset has a total of 26 304 hourly observations. Weather is a dataset [1] that contains local climate data from Germany, across three years. The datasets ETT, ECL, and Weather all natively have a sub-day resolution which we aggregate into daily observations by taking the sum of values in day.

### E.5 Noise Removal Ablation

Recall that crucially, since we use synthetic training data with a ground-truth trend and a separate noise component, we remove the noise when calculating the train loss. Now, we give an ablation study to confirm that this design decision significantly improves the performance of our model.

We investigate the impact of this design decision on the performance of ForecastPFN by comparing the model with the noise removed (that which is included in the main body of the paper) with a training run conducted without the removal of the noise from the training loss calculation. In doing this, we calculate the validation MSE on the synthetic data for both models and observe **33.089** for the model without noise removed and **22.692** for the model with the noise removed. We conclude the use of the noise removal during training loss calculation leads to improvements on the synthetic validation set, and thus we deploy the use of noise removal in the final model.

This design decision significantly improves performance, and it *can only be done when using synthetic data*, rather than real data.

### E.6 Training Time Analysis

Recall that during our benchmark, there were six models which required training data on a portion of each of the datasets before they were able to be tested. Further recall that we established a time budget for this training process, and varied that timing from 1 to 120 seconds. Some of these models were unable to fit to the trainig data during some of those training budgets. In Table 3, we calculate the percentage of configurations (dataset, prediction lengths, seeds) which failed for each model and training time budget. We see that Arima is the model which struggles the most at low time budgets.

### E.7 Additional Experimental results

In Figure 16 and Figure 6, we plot the mean MSE rank per prediction length (similar to Figure 3 and 4 respectively, but for mean MSE rank instead of MSE wins).

In Figures 8 to 20, we give similar plots to Figures 3, 4 and 6 but instead of MSE, we plot MAE, MAPE, and MSPE. We also include plots for the evaluation of mean rank.

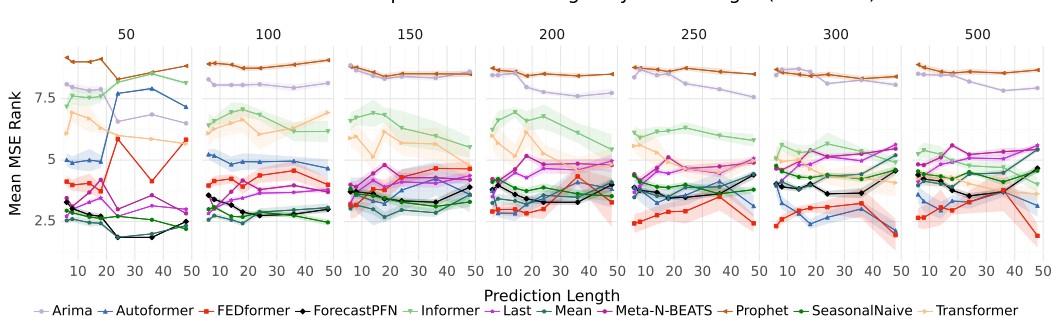

Figure 6: Analysis of performance across prediction lengths for increasing data budgets (left subplot, data budget is 50 to the right subplot, data budget is 500), aggregated across datasets. Mean MSE rank for each scenario is plotted with error bars as one standard deviation across training runs. Recall, ForecastPFN and Meta-N-BEATS see no training data for these series, only the length 36 input.

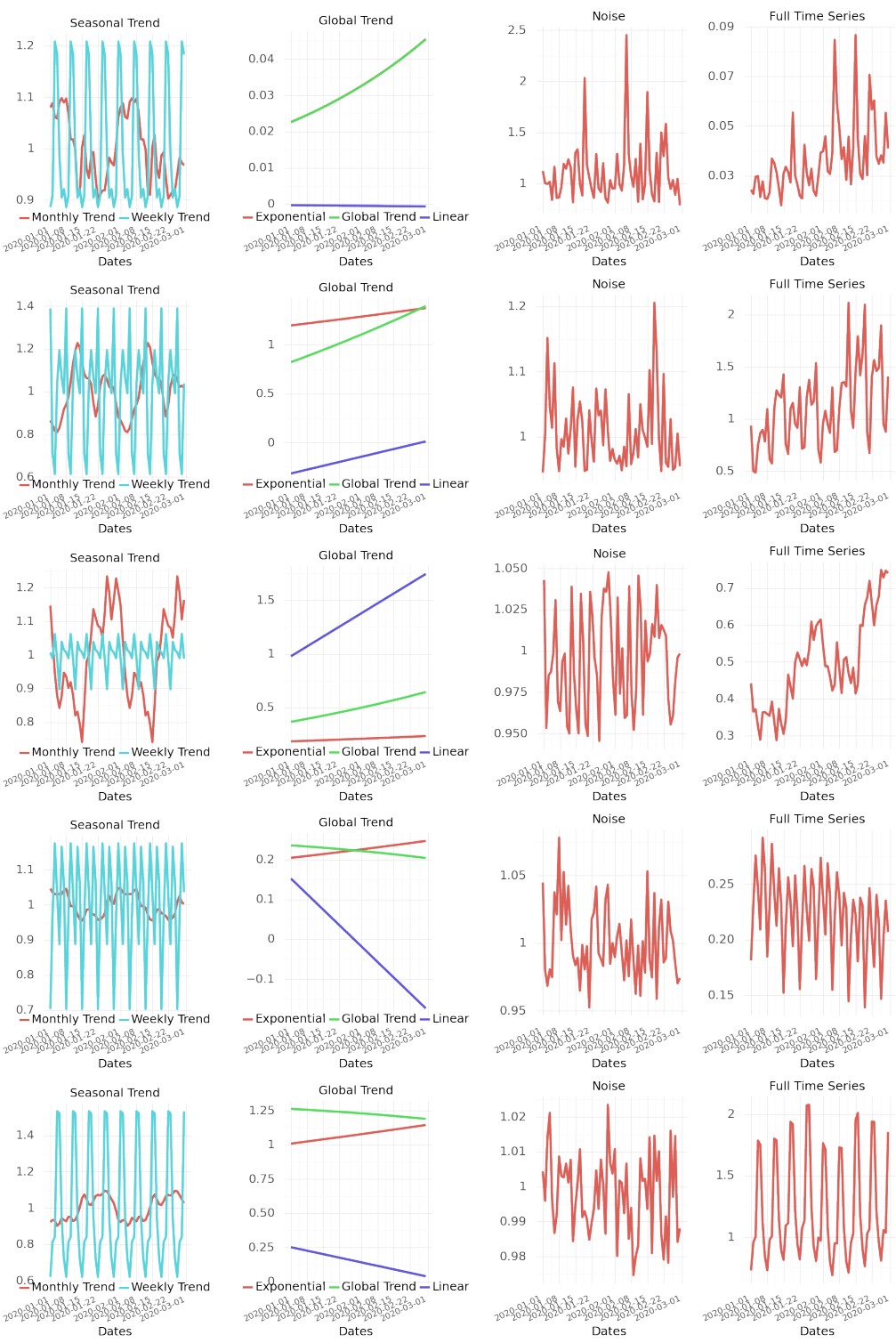

Figure 7: Five examples of time series drawn from the synthetic distribution we define in Section 3.2 (continuation of Figure 2). Each row shows the components that make up a time series: seasonal trends (far left), global trends (center left), and noise (center right), as well as the entire time series (far right). We plot only half of the series in order to show the seasonal trends clearly.

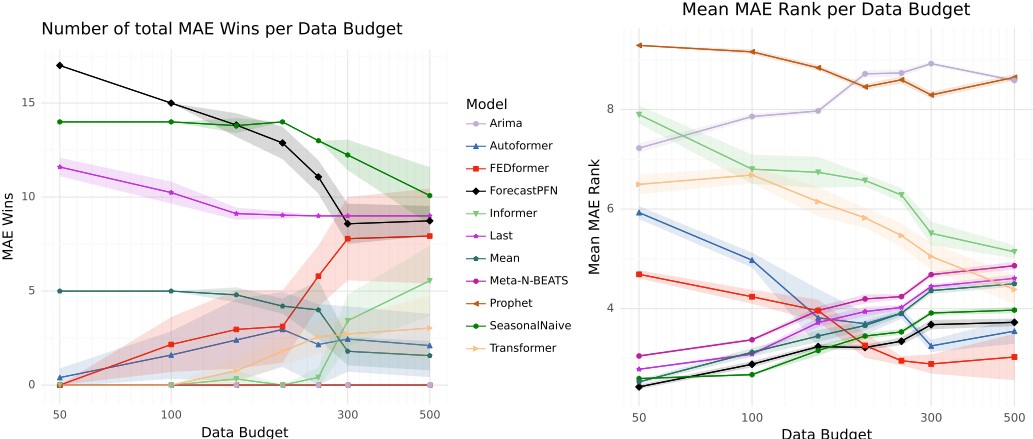

Figure 8: Analysis of performance vs. **data budget**, aggregated across datasets and prediction lengths. We plot the number of total MAE wins (left) where a higher value is better and mean MAE rank (right) where a lower values is better. Error bars show one standard deviation across training runs. Recall, ForecastPFN and Meta-N-BEATS see no training data for these series, only the length 36 input.

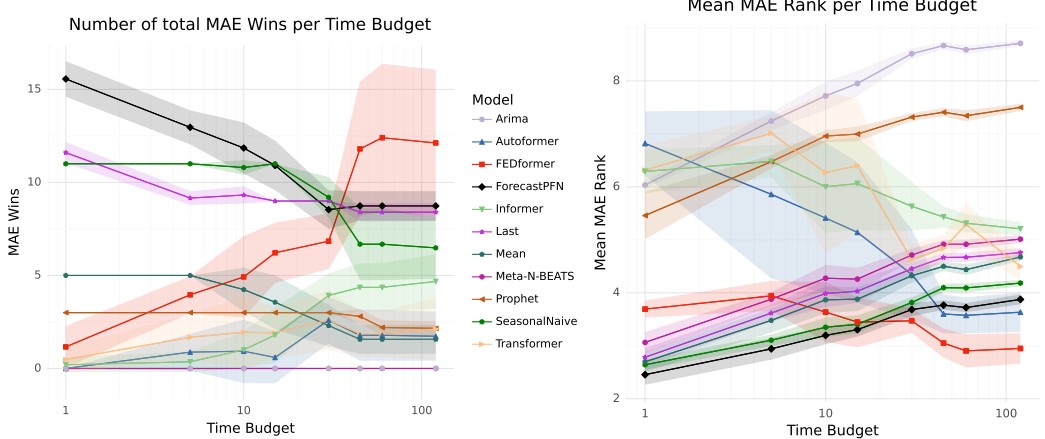

Figure 9: Analysis of performance across **time budgets**, aggregated across datasets and prediction lengths. We plot the number of total MAE wins (left) where a higher value is better and mean MAE rank (right) where a lower values is better. Error bars show one standard deviation across training runs. Recall, ForecastPFN and Meta-N-BEATS see no training data for these series, only the length 36 input.

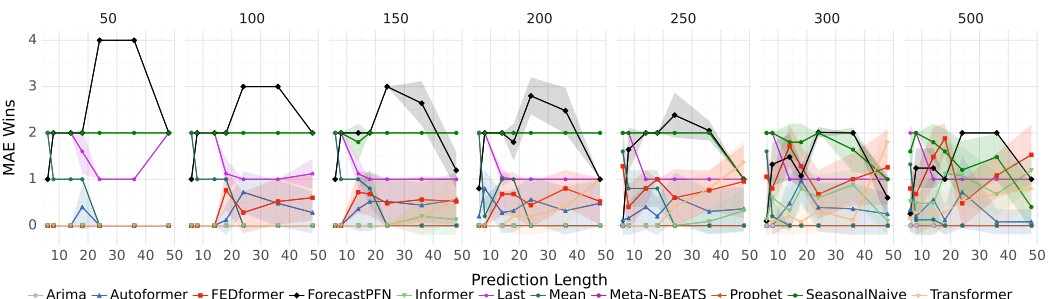

Figure 10: Analysis of performance across prediction lengths for increasing data budgets (left subplot, data budget is 50 to the right subplot, data budget is 500), aggregated across datasets. Total MAE wins for each scenario is plotted with error bars as one standard deviation across training runs. Recall, ForecastPFN and Meta-N-BEATS see no training data for these series, only the length 36 input.

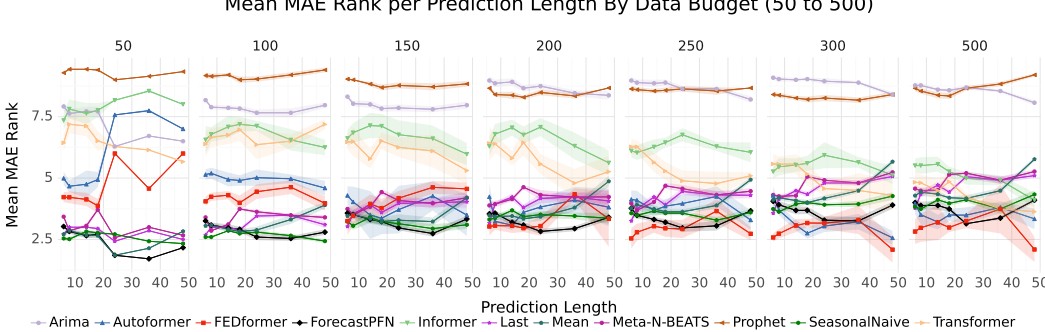

Figure 11: Analysis of performance across prediction lengths for increasing data budgets (left subplot, data budget is 50 to the right subplot, data budget is 500), aggregated across datasets. Mean MAE rank for each scenario is plotted with error bars as one standard deviation across training runs. Recall, ForecastPFN and Meta-N-BEATS see no training data for these series, only the length 36 input.

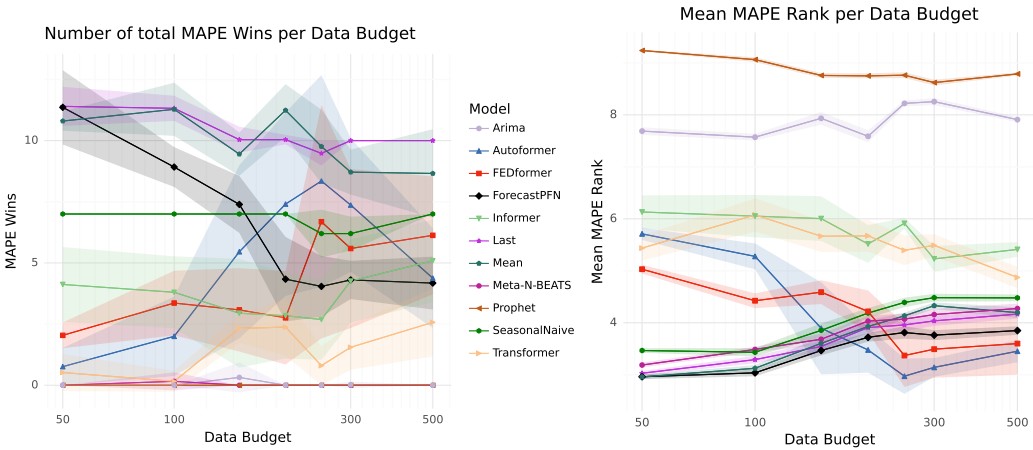

Figure 12: Analysis of performance vs. **data budget**, aggregated across datasets and prediction lengths. We plot the number of total MAPE wins (left) where a higher value is better and mean MAPE rank (right) where a lower values is better. Error bars show one standard deviation across training runs. Recall, ForecastPFN and Meta-N-BEATS see no training data for these series, only the length 36 input.

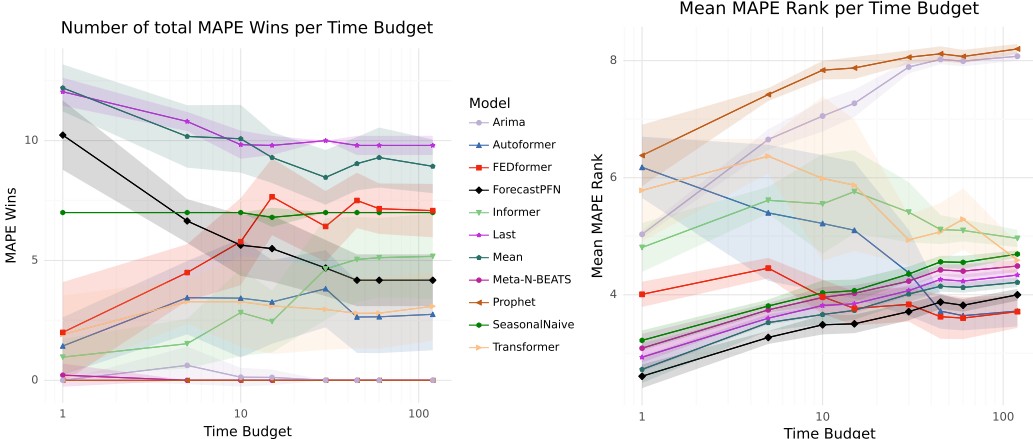

Figure 13: Analysis of performance across **time budgets**, aggregated across datasets and prediction lengths. We plot the number of total MAPE wins (left) where a higher value is better and mean MAPE rank (right) where a lower values is better. Error bars show one standard deviation across training runs. Recall, ForecastPFN and Meta-N-BEATS see no training data for these series, only the length 36 input.

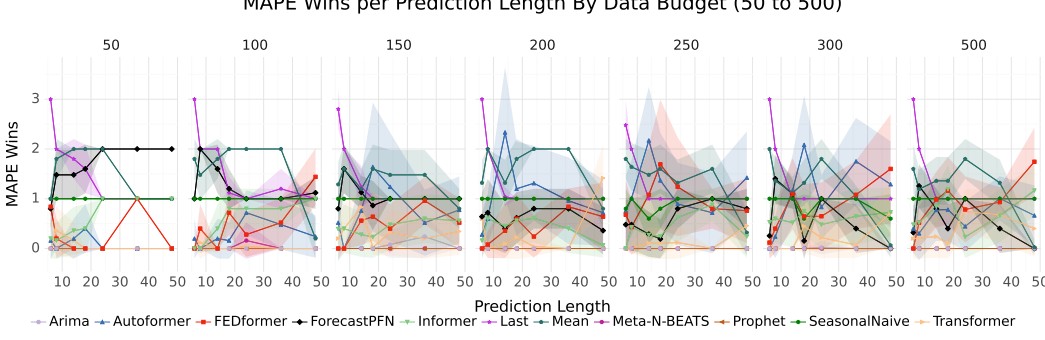

Figure 14: Analysis of performance across prediction lengths for increasing data budgets (left subplot, data budget is 50 to the right subplot, data budget is 500), aggregated across datasets. Total MAPE wins for each scenario is plotted with error bars as one standard deviation across training runs. Recall, ForecastPFN and Meta-N-BEATS see no training data for these series, only the length 36 input.

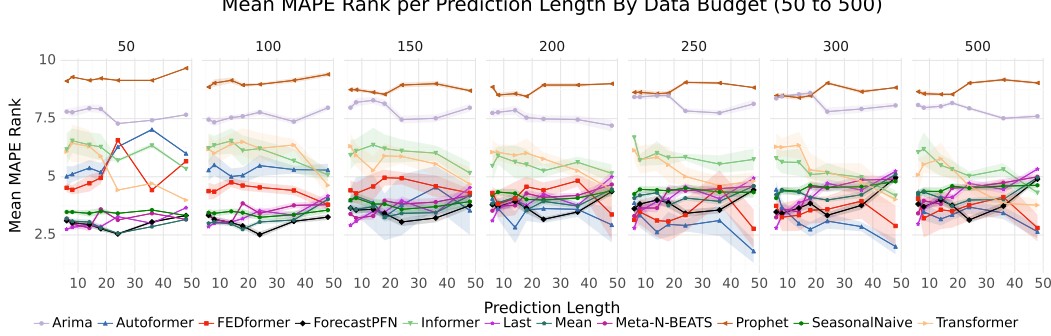

Figure 15: Analysis of performance across prediction lengths for increasing data budgets (left subplot, data budget is 50 to the right subplot, data budget is 500), aggregated across datasets. Mean MAPE rank for each scenario is plotted with error bars as one standard deviation across training runs. Recall, ForecastPFN and Meta-N-BEATS see no training data for these series, only the length 36 input.

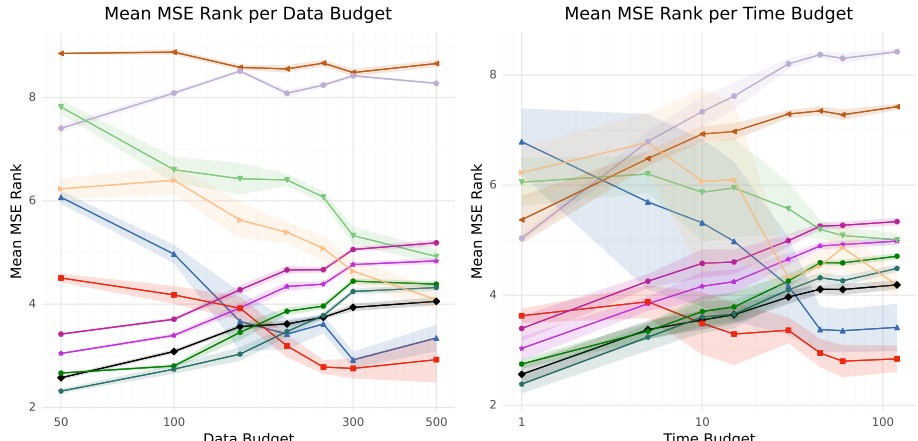

Figure 16: Analysis of performance vs. **data budget** and **time budget**, aggregated across datasets and prediction lengths. We plot the number of total MSE wins (left) where a higher value is better. Error bars show one standard deviation across training runs. Recall, ForecastPFN and Meta-N-BEATS see no training data for these series, only the length 36 input.

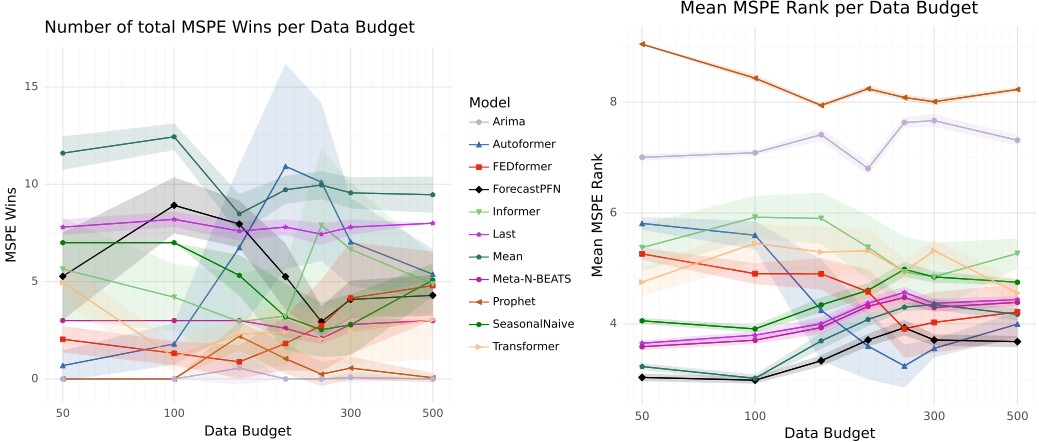

Figure 17: Analysis of performance vs. **data budget**, aggregated across datasets and prediction lengths. We plot the number of total MSPE wins (left) where a higher value is better and mean MSPE rank (right) where a lower values is better. Error bars show one standard deviation across training runs. Recall, ForecastPFN and Meta-N-BEATS see no training data for these series, only the length 36 input.

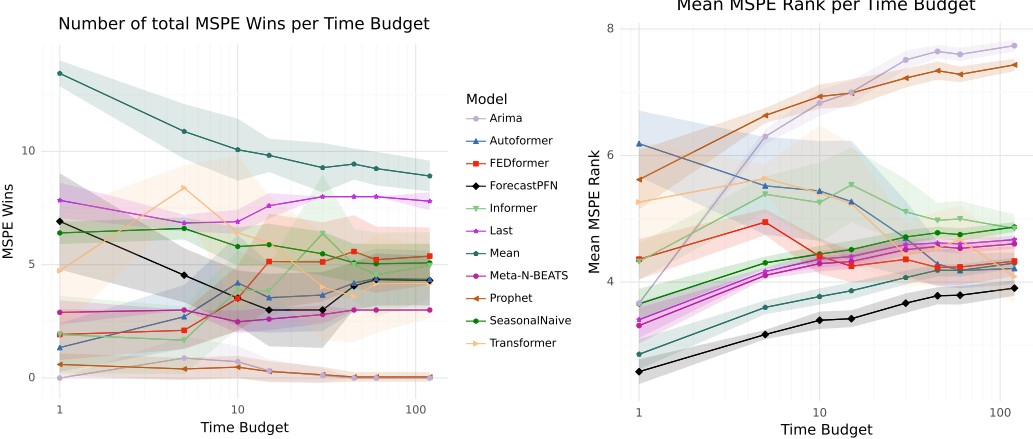

Figure 18: Analysis of performance across **time budgets**, aggregated across datasets and prediction lengths. We plot the number of total MSPE wins (left) where a higher value is better and mean MSPE rank (right) where a lower values is better. Error bars show one standard deviation across training runs. Recall, ForecastPFN and Meta-N-BEATS see no training data for these series, only the length 36 input.

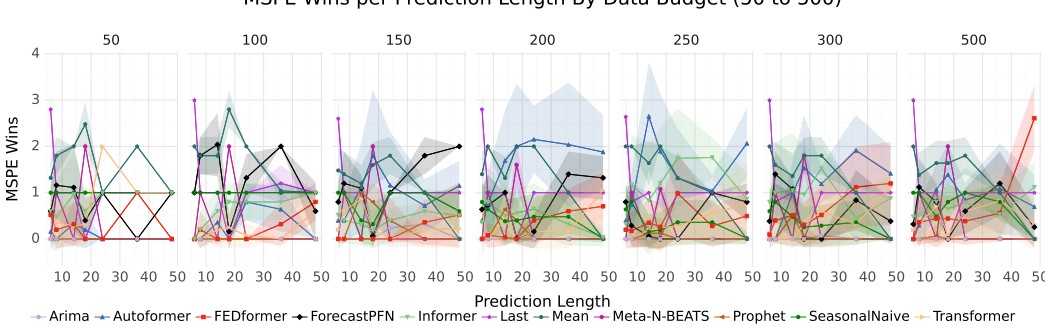

Figure 19: Analysis of performance across prediction lengths for increasing data budgets (left subplot, data budget is 50 to the right subplot, data budget is 500), aggregated across datasets. Total MSPE wins for each scenario is plotted with error bars as one standard deviation across training runs. Recall, ForecastPFN and Meta-N-BEATS see no training data for these series, only the length 36 input.

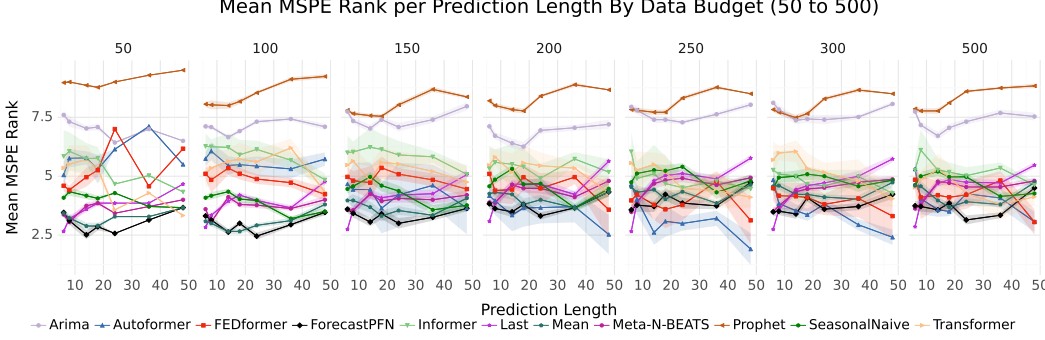

Figure 20: Analysis of performance across prediction lengths for increasing data budgets (left subplot, data budget is 50 to the right subplot, data budget is 500), aggregated across datasets. Mean MSPE rank for each scenario is plotted with error bars as one standard deviation across training runs. Recall, ForecastPFN and Meta-N-BEATS see no training data for these series, only the length 36 input.

