# OpenReview forum: "ForecastPFN: Synthetically-Trained Zero-Shot Forecasting"
_NeurIPS.cc/2023/Conference — NeurIPS 2023 poster_

### Official Review · Reviewer_458V · 2023-06-24

**Soundness:** 3 good
**Presentation:** 3 good
**Contribution:** 2 fair
**Rating:** 6
**Confidence:** 4

**Summary:**

The work present the ForecastPFG a zero-shot forecasting method trained using only synthetic dataset and it is evaluated on several real world dataset.

**Strengths:**

The paper is well written and the steps well described.
Moreover, I think that it could be an interesting approach when you have very few data.

**Weaknesses:**

Weakness are inserted in the Question section

**Questions:**

Rows 71-74: This a repetition of what authors wrote in rows 16-20. This part should be improved with some details. The same in rows 83-85 and rows 23-24.

The relation between Zero shot forecasting, Transfer Learning in the forecasting context and the concept of Global Models is not discussed properly. I suggest authors to discuss more about this relation.
Regarding global models authors can refer to recent works as: Januschowski et al., "Criteria for classifying forecasting methods", 2020; Buonanno et al., "Global vs. Local Models for Short-Term Electricity Demand Prediction in a Residential/Lodging Scenario", 2022; Montero-Manso, P.; Hyndman, R.J. "Principles and algorithms for forecasting groups of time series: Locality and globality", 2021, etc.

Rows 102-104
Authors use a multiplicative decomposition ($y_t = \gamma(t) \cdot z_t$). Why this choice and not the additive decomposition ($y_t = \gamma(t) + z_t$)? How this choice impact on the results of the work?

Row 109-110
Could you clarify this tense? The $y_t = \gamma(t) \cdot z_t$ is always valid not only when $\gamma(t)$ is deterministic. Moreover, if it is present $z_t$ sampled by a Gaussian, there are different possible output values and not only $y_T$. Do I miss something?

Rows 113-114
The athors wrote:
"We optimize the PFN’s parameters $\theta$ by making predictions on a held-out test set from D"--> Usually with test set is named the set used exclusively for the evaluation and that is not involved in learning of the model. In this case authors define a test set used to evaluate the loss used for the parameter tuning. I think that the name is misleading. Authors can evaluate to use different names (e.g. Training Input and Training Output) to identify the two parts of the timeseries used in the training phase.

Rows 129.
Is not crucial to know the prior distribution of timeseries in order to have a good result on real dataset? In this work authors use a large dataset of timeseries that try to cover different situations but how the choice of the prior distribution impact on the results? Maybe this can be a idea for future works.

Rows 144
Is there a motivation on the usage of Weibull distribution for the noise?
How a choice of a different noise distribution impacts on the results?

Rows 146-147
Now $\gamma(t)$ contains two contributions: trend and seasonal. In the row 102, instead, is written only trend. I got that in 102 authors want to introduce the general frame of the problem but I would not use the name trend in row 102.

Rows 169
"Predict the MSE of this query" --> Does not the architecture predict the output? The next rows in fact the authors wrote "the output is a single prediction at a future time".
Also at row 178 it is written "the loss" and not the output.

Rows 180-183
It is not clear to me why the classic scaling techniques are not appliable in this case. As discussed after, the problem is the outlier that has to be excluded since they can be very different across several timeseries. Please express better this issue.

Rows 188-191
Is MSPE the mean squared prediction error?
Could you describe mathematically the issue raised in this section because I didn't get it.

Row 196
How authors evaluate the convergence?

Row 235-236
Not always the forecasting models are evaluated in retraining mode. For having a fairer comparison authors could force the forecasting model to not retrain during the evaluation test. This condition is, in fact, more similar to what happen in the real context where you don't retrain continuously your model once it is deployed. Moreover, this modality (no retrain) is followed in several works about forecasting. Using this modality (no retrain) we should have a fairer comparison between ForecastPFN and other models.

Row 255
Cross product --> do you mean cartesian product?

Figure 4
Is the Time Budget the same of wall clock budgets defined in row 243

Row 301
Is not more important to compare classic standardization (without outlier removal) with robust scaling? The authors, instead, compared robust scaling with min-max scaling.

**Limitations:**

A limitation I see is the maximum number of training data used for the other models, limited to 500. It looks clear that with more other models outperform the ForecastPFN (Fig.5) while with few data ForecastPFN looks a good solution.
Moreover, the training of ForecastPFN requires 30hr so it has a cost that should be taken in account for having a fair comparison with other approaches.

---

> ### Author Rebuttal · Authors · 2023-08-10
>
> Thank you for the excellent and detailed feedback. We are glad to see your positive view of our work. Your comments and suggestions will greatly improve the final version of our paper. We reply to each question below.
>
> **Q1 Repetition of text.**
> Thank you for catching this. We have now added more details to lines 71-74 and 83-85.
>
> **Q2: Relation to transfer learning and global models.**
> Thank you for pointing this out. We will add a section on transfer learning, and another section on global models to our related work section.
>
> **Q3: Multiplicative decomposition.**
> Yes, additive vs. multiplicative noise is an interesting question. We chose multiplicative noise to better balance the amount of signal to noise across all series. If we used additive noise, then the impact of the noise depends on the ratio of the scale of the base series and the scale of the noise. Since our base series have linear and exponential terms, additive noise would cause the signal-to-noise ratio to vary substantially over series with a trend component. Therefore, we used multiplicative noise, which is a simpler parametrization to achieve a signal to noise ratio for a series. We now updated our paper to include this discussion, and we agree that including an ablation study is a good idea for future work.
>
> **Q4 Clarify equations.**
> Thank you for pointing this out. This is indeed a typo, because $p(y_T|T,\varphi)=1$ is only true if there is a deterministic function $y_t=\varphi(t)$ with no noise, and this is not the case we consider in Section 3.2.
>
> **Q5 Clarification on test set.**
> Thank you for catching this! As you say, this is simply a naming mistake. We will now refer to the two parts of each training series as “training input” and “training output.”
>
> **Q6 Ablation on prior distribution.**
> We agree that the choice of prior distribution is important and that it would be good to have an ablation study. It is challenging to do a thorough ablation study, since it takes 30 GPU hours to train a model. As mentioned in the paper, we did light tuning in the early stages of the project, using the training loss. We are currently working on a simple ablation study on the noise of the synthetic data, which we will finish during the NeurIPS discussion period.
>
> **Q7 Weibull distribution.**
> This is a great question. Real world datasets seem to frequently have two types of noise: Gaussian and exponential; the latter is in the form of non-negative time series with high value outliers. The Weibull distribution is a simple and natural way to parametrize between these two distributions. While we do not claim the Weibull distribution is representative of the noise of all real world series, getting convergence on Weibull is already particularly non-trivial. Further study of the noise distribution is an interesting idea for future work.
>
> **Q8 Naming of phi.**
> We agree that it is clearer to replace “trend” in line 102 with a different phrase, such as “base series.”
>
> **Q9 Prediction.**
> Thanks for pointing this typo on rows 169 and 178. Yes, the architecture predicts the output, and then we compute the MSE of the output with the ground truth.
>
> **Q10 Classic scaling techniques.**
> Yes, we will make this part clearer in the paper. Since ForecastPFN is designed to be a universal forecasting model that can handle a huge variety of time series, appropriately scaling the data becomes tricky. Our robust scaling procedure as described on line 184 gives the best level of standardization and outlier removal, which simpler methods such as min-max scaling or z-score normalization do not achieve.
>
> **Q11 MSPE.**
> By MSPE, we are referring to the [mean squared percentage error](https://www.sktime.net/en/stable/api_reference/auto_generated/sktime.performance_metrics.forecasting.mean_squared_percentage_error.html).
> It is a great idea to give the equations of MSE, MSPE, and our scaled MSE. We will make this change to the final version of our paper.
>
> **Q12 Convergence.**
> We used the word convergence informally to mean that the training loss is much higher than the final trained ForecastPFN. So, if we set the noise of the synthetic data too high, the training loss is very high. We will fix this in the text.
>
> **Q13 Retraining.**
> Thank you for pointing out this confusion. By “not a fair comparison”, we meant that the baseline methods are allowed to train on earlier time series from the same dataset. At test time, we do not permit the baseline methods to train on the test input. Indeed, we do the common technique of training the algorithms once per dataset and prediction length, and then, at test time, providing input to the model for $n$ timestamps and then asking the model to predict $l$ points in the future. We have updated the manuscript to clear up this confusion.
>
> **Q14 Cross product.**
> Yes, in row 255 we meant to say the cartesian product. Thank you for spotting this.
>
> **Q15 Figure 4.**
> Yes, “training time” in row 243 is the same as “time budget” in Fig. 4. We will make this clear in the paper.
>
> **Q16 Standardization.**
> That is a good point. In future work, we can do a much larger ablation study for scaling methods and the data distribution. It is challenging to do a large ablation study because each ForecastPFN model takes 30 GPU hours of pretraining.
>
> Thank you very much, once again, for your positive view of our work and your excellent suggestions. If you find our responses satisfying, we respectfully ask that you consider increasing your score. On the other hand, we would be very happy to answer any follow-up or additional questions you have.

---

> > ### Comment · Reviewer_458V · 2023-08-19
> >
> > Thanks for the answers and clarifications. I decided to increase the my evaluation.

---

### Official Review · Reviewer_3SYZ · 2023-07-01

**Soundness:** 1 poor
**Presentation:** 3 good
**Contribution:** 3 good
**Rating:** 5
**Confidence:** 3

**Summary:**

The paper proposes to pre-train a deep learning model on synthetic data which follows characteristics of real world time series data. This model can then be used for any downstream forecasting dataset. The authors performed experiments to show that the proposed method performs well compared to existing classical and deep forecasting models in constrained scenarios (restricted number of training data, restricted amount of time for training).

**Strengths:**

The paper proposes an exciting direction of pre-training deep forecasting models with synthetic data, which can be quickly adapted in a Bayesian manner on a new unseen real time series, and be able to perform accurate forecasts. The synthetic data is proposed using a prior distribution which has properties similar to real world data. Such a direction is exciting and of huge significance, unlocking the power of deep learning models to be pre-trained on large datasets. The paper is well written, neatly organised in a logical flow, and contains comprehensive experiments.

**Weaknesses:**

* While the premise of the paper is indeed exciting, I believe the claims made in the abstract and introduction are greatly exaggerated. Claims are made that the proposed method, ForecastPFN is able to beat SOTA forecasting methods. However, the experiments are only performed on severely handicapped scenarios (SOTA models are only given access to 100s of data points or maximum of 2 minutes of training). More effort should be made to highlight this handicapped scenario.
* Claims that "the zero shot methods are at a disadvantage because the other six methods were allowed to train on data from the  evaluation time series" (lines 280 - 282) seem a little disingenuous since the zero shot methods were pre-trained extensively (30 hours of pre-training for ForecastPFN), whereas in certain cases the six methods were only allowed 1 second of training?
* The claim in line 173 - 175, that existing transformer models have prediction length to be fixed is inaccurate. This is a feature of Transformers in the long sequence time series forecasting setting, but not true of time series Transformers in general. See [1] for one such example.
* Only number of wins/ranking is given for the results, but not the actual MSE scores. The actual MSE values should be given for readers to understand whether the comparisons are meaningful at all.
* Following up on the previous point, 2 more very simple baselines should be added - the naive forecast (current value is taken as the forecast) and the historical mean. The number of wins/rankings doesn't really matter if any of these models can't beat such naive baselines (which is a fair comparison regardless of the train/time budget).

[1] Rasul, Kashif, et al. "VQ-TR: Vector Quantized Attention for Time Series Forecasting." (2022).

**Questions:**

1. Are the existing methods (FEDformer, etc.) also pre-trained on the synthetic data? If not, it seems quite artificial to claim wins when these methods are only given 50 datapoints (1 batch?) for training, or 1 second to train. In fact, it seems unfair to compare to ForecastPFN which has 30 hours of pre-training on a huge synthetic dataset.
2. How was Meta-N-BEATS trained on the M4 dataset? In the paper, 1 model was trained for each frequency. Which subset of M4 was the baseline trained on?
3. What does number of MSE wins mean?
4. What does training on 1 second mean? How is this implemented? Was a single gradient step even performed in this setting?
5. What are the experiment details of the baseline models? What machine was used to run these baseline experiments? It seems a little iffy to compare the time budget in terms of seconds. How to compare 120 seconds on a H100 GPU vs a K80 GPU or even a CPU?
6. How was the time restricted training for ARIMA/Prophet implemented? I understand for deep learning models you can easily stop the training, but what about ARIMA/Prophet if the model has not been fitted?

Please consider using number of batches/epochs trained instead of wall clock seconds.

**Limitations:**

The paper doesn't adequately address limitations. The limitations given in the limitations section aren't really true limitations, but rather things that are out of scope of this paper. Some suggestions of limitations to consider:
1. What if the proposed prior for synthetic data is completely different from the downstream real world data? Perhaps some sort of adversarial forecasting task?
2. ForecastPFN doesn't compute the posterior predictive distribution, it is just an approximation.
3. Does not consider probabilistic forecasting, which has more implications on the synthetic prior (not just the shape of time series, but also the distribution of data, need to consider some autocorrelation in the errors)

---

> ### Author Rebuttal · Authors · 2023-08-10
>
> Thank you for the excellent feedback. We are very glad to see that you view this direction as exciting and of huge significance, and that you found the paper well-written, neatly organized, and with comprehensive experiments. We very much appreciate your comments, which we believe will substantially improve the final version of our paper. We reply to each question below.
>
> **W1-2: “handicapped scenario” / “disingenuous claims.”**
> Thank you for bringing up these points. We understand your points and have now made the experimental setup and settings very clear in our paper. We would like to bring up a few points for consideration. **The crux of this issue is that we focused on a novel paradigm that diverges significantly from prior work:** zero-shot forecasting; pretraining; and a setting that is important in practice: very limited in-distribution data and/or inference time. This makes it challenging to compare to prior work in general. To be clear, ForecastPFN is trained only once ever, and the weights are never changed when evaluating on new, unseen tasks. On the other hand, FEDFormer and the other transformers were not designed to be pretrained or to run zero-shot predictions. However, based on your comments, we *do* try out pretraining for FEDFormer. We find that after a week of attempting to train FEDFormer on the synthetic data, we were unable to achieve any non-degenerate performance on the 7 real-world datasets. Note that even training the ForecastPFN architecture (designed to be flexible and universal) was non-trivial: as described in the paper, we faced technical challenges such as handling scaling robustly, and a delicate parameterization of the synthetic data parameters.
>
> Overall, our work introduces a new paradigm in which prior works cannot easily fit, and our model also performs particularly well in an important setting -- the low resource setting, for which most prior work was not designed. We are now much more explicit with our claims by mentioning the exact training settings, and noting how these differ from the settings for which prior work was developed. Thank you very much for giving us the opportunity to further clarify these points in our paper.
>
> **W3. Transformer prediction length.**
> Thank you for catching this! We have now corrected it.
>
> **W4. Raw MSE values.**
> Thanks for this suggestion; we can easily add the raw MSE values. See [our reply to Reviewer MwwS](https://openreview.net/forum?id=tScBQRNgjk&noteId=Z6mb79DbuS) and also the one-page pdf.
>
> **W5. Add two additional baselines.**
> We really like this suggestion. We added these two baselines; see the one-page pdf. We find that the historical mean achieves second place behind ForecastPFN in the lowest training and runtime settings. This is not unexpected and is a testament to the challenge of predicting a series when given very little training data or training time.
>
> **Q1. FEDFormer and other models were not pretrained.**
> Thank you for this suggestion. We give our full answer in the first bullet point above: in general it is non-trivial to achieve non-degenerate zero-shot performance (even for specially designed architectures such as ForecastPFN), and we have not been able to do so for FEDFormer after a week of effort.
>
> **Q2. Meta-N-Beats setup**
> Meta-N-Beats was trained separately for each configuration of prediction length and dataset frequency, just like the other Transformer baselines. We use the subset of M4 according to the corresponding frequency. In contrast, the more flexible ForecastPFN can accommodate different frequencies and prediction lengths as a single model.
>
> **Q3 What is “number of MSE wins.”
> This means that we compute the MSE for a given configuration and count the number of times each model has the best MSE. For example, in Figure 3, we count the number of wins for each model across 7 datasets and 7 prediction lengths, for each train budget.
>
> **Q4&6 Training on 1 second / Constrained budget for ARIMA/Prophet.**
> Thank you, we will explain this better in the paper. Following the experimental setup of [TabPFN](https://arxiv.org/abs/2207.01848), we give each training run a time budget. After each training step, we stop the training if the total budget was exceeded. In case an algorithm is unable to output any predictions within the train budget (such as ARIMA with a 1 second budget), we set the output to 0’s.
>
> **Q5 Hardware used.**
> We ran all the experiments ourselves on a V100; therefore, all of our results are fair to compare. We respectfully point out that since we include a diverse set of methods, the epoch/step times are not comparable, and so the best practice is to report the wallclock time using the exact same hardware, as we did. For example, [footnote 8 here](https://arxiv.org/abs/1909.02453) has a good discussion. We would be happy to include other compute resource metrics in the appendix while being explicit about the aforementioned caveats.
>
> **Limitations.**
> Thank you for these great suggestions. We agree; our synthetic prior was created to focus on “human-like” or “earth-like” time series, but it would not do well on a purely “mathematical” time series with uncommon periods such as 41, 89, or 97. We also will clarify that our method only approximates the posterior predictive distribution and makes pointwise predictions. Since this is a new type of work, we focused on the pointwise predictions, but probabilistic predictions are an exciting area for future work.
>
> Thank you very much, once again, for your excellent feedback. Your perspective and comments are very important for us to properly convey our work. We hope that you start to see our perspective on the main issues. Please let us know if you have any follow-up comments. If this starts to address your questions, especially since you agree that this direction is exciting and significant, we hope that you might reconsider giving our work a “strong reject” rating; we would very much appreciate it if you consider increasing your score.

---

> > ### Comment · Reviewer_3SYZ · 2023-08-13
> >
> > Thank you authors for the time and effort in crafting an extensive rebuttal. Overall I still have major concerns regarding the fairness of the comparisons.
> >
> > 1. I am still confused what is the point of comparing the settings for 1 - 10 seconds of training. Loading the data probably takes more than 1 second?
> >
> > 2. Why not let the fallback forecast when the model fails to output any predictions be the naive forecast?
> >
> > 3. There is also another simple baseline called the seasonal forecast. https://otexts.com/fpp2/simple-methods.html
> >
> > 4. Can we get some indication on how often in your plots, does ARIMA and Prophet fail to output a prediction?
> >
> > 5. I am wondering how realistic / in what real world settings is the case of having a limited time budget. It may be the case for ARIMA, whereby you usually would want to retrain the model each time new observations come in. But for deep learning models, usually we train the model once, and can use it for a long time, i.e. no need to train the model again when new observations come in.
> >
> > 6. I agree that number of steps/epochs is not the right measure of compute resources in this case, since the setting being explored is when time is a constrain. However, wall clock is still not a reliable measure. There are so many unknowns as a reader - what was the batch size picked? Was the most efficient batch size picked for all methods? Was the most efficient implementation of ARIMA used? Does Prophet have a GPU implementation? I find it hard to draw any conclusions from the results without knowing any of these details. FLOPs seems to be a much more suitable measure of compute resource here, as it allows us to ignore all these questions about efficiency of implementation.
> >
> > 7. You mentioned that the ForecastPFN architecture is non-trivial. I am a little confused regarding this - based on my understanding, the architecture is a standard Transformer. Apart from the synthetic data generation and some scaling pre-processing, what is new in the architecture?

---

> > > ### Author Response · Authors · 2023-08-19
> > > **Second reply to 3SYZ, part 1**
> > >
> > > Thank you very much for replying and giving us a chance to further clarify and improve our submission. Overall, we are happy to see that your new questions are relatively minor, such as correctly handling low-runtime baseline predictions, clarifying the metrics, and discussing real-world applications for the low-resource setting, and we especially appreciate your pointer to the new baseline, which we have now added.
> > >
> > > **1. Data loading time.**
> > > Loading training data can take longer than 1 second. However, our training time, as in the TabPFN paper, only starts the runtime after the data and model are loaded and gradient computations start. See #5 for motivation for why it is important to evaluate 1-10 second of training.
> > >
> > > **2. Fallback forecasts when the model fails.**
> > > After further thought, we believe that the best way to handle this issue is to do the same thing as the original [TabPFN paper](https://arxiv.org/abs/2207.01848) (Fig. 5): for each baseline, start showing results when the baseline is able to make non-trivial predictions on **all** datasets/settings. We will also include raw results for each dataset for completeness. This is the best way to update our paper so as not to cause confusion about the low-runtime setting, and it is backed up by existing work. (For the specific runtimes of baselines, see answer #4.)
> > >
> > > You bring up a great point that in real life, if using a model such as FEDFormer, then the practitioner would use FEDFormer for high budgets and the seasonal baseline for low budgets. The performance of this combination can easily be inferred from our plots by looking at the two corresponding methods, and we will make a point of this as a comparison to ForecastPFN in our updated manuscript.
> > >
> > > **3. Seasonal baseline.**
> > > Thank you, we are happy to add more baselines. Many of our low-resource experiments involve series that are fewer than 365 days long, so instead of monthly we use weekly, which uses the historical mean from the same day of the week as the current prediction. This baseline now becomes the 2nd best method in low train size and low runtime settings, 2nd to ForecastPFN, across all axes (MSE wins across train and time budgets, mean MSE rank across train and time budgets). See part 3 of our reply for these results. We have updated all plots in our paper to include this baseline.
> > >
> > > **4. ARIMA and Prophet minimum times.**
> > > We have calculated the percentage of configurations (dataset, prediction lengths, seeds) that failed, for each time budget and each model which requires training or being fit. We see that Arima is the model which struggles the most at low time budgets.
> > >
> > > | Model/Time Budget       |   1.0 |    5.0 |   10.0 |   15.0 |   30.0 |   45.0 |   60.0 |   120.0 |
> > > |:------------|------:|-------:|-------:|-------:|-------:|-------:|-------:|--------:|
> > > | Arima       |  9.68 |  77.42 |  88.71 |   92.9 | 100    | 100    | 100    |     100 |
> > > | Autoformer  | 70.97 |  97.74 | 100    |  100   | 100    | 100    | 100    |     100 |
> > > | FEDformer   | 71.94 |  99.68 | 100    |  100   | 100    |  99.35 | 100    |     100 |
> > > | Informer    | 72.58 | 100    | 100    |  100   | 100    | 100    |  97.42 |     100 |
> > > | Prophet     | 88.71 | 100    | 100    |  100   | 100    | 100    | 100    |     100 |
> > > | Transformer | 66.77 | 100    | 100    |  100   |  98.39 | 100    | 100    |     100 |

---

> > > ### Author Response · Authors · 2023-08-19
> > > **Second reply to 3SYZ, part 2**
> > >
> > > **5. Motivation for the low runtime setting.**
> > > First, we emphasize that there is a wide range of applications for a limited training set size, as cited in our paper (lines 23-24, 83-88).
> > >
> > > Next, there is also a wide range of applications which require a low time (or computational) budget. As you say, deep learning models can train once and be used for a long time for in-distribution data. But, they will need to be retrained for new applications (out-of-distribution), unlike ForecastPFN. As an example of a nice property, ForecastPFN can be loaded onto a CPU (a standard function of tensorflow) and used on edge devices to make predictions. As an example of a specific application, ForecastPFN can then be used to forecast personalized utilities usage across households and across towns in developing countries. Other deep learning models would need to retrain when the data becomes out of distribution. Another application is as a forecasting data-exploration tool, allowing users to see *instant* forecasting visualizations as they navigate their dataset.
> > >
> > > **6. Runtime.**
> > > We respectfully disagree with your statement that “wall clock is still not a reliable measure”.
> > > - First, recall that on lines 218-220, we mentioned that all of the baselines use the hyperparameters from their official release, and we specified the ARIMA implementation. We also open-sourced our entire codebase (line 61), so that readers can see the exact code and reproduce experiments if they want.
> > > - Second, there is an enormous precedent throughout the ML literature to use wallclock time as a metric. Wallclock time is used in many prior works and is mentioned in best practices guides.
> > >
> > > We do agree that we will specify the fact that ARIMA and Prophet do not use GPUs, and the CPU version (it's an N1 series from GCP). We are happy to include additional plots that use FLOPs instead of wallclock time, but we will need to add caveats that FLOPs can sometimes be misleading. As a quick example, a vanilla transformer is highly GPU-memory bottlenecked, and would appear to be slower in FLOPs than an LSTM, even though LSTMs are substantially slower in wall clock time. (This issue is solved by using Flash Attention for transformers).
> > >
> > > **7. ForecastPFN architecture.**
> > > The transformer is more flexible than existing transformers for forecasting, due to the input that can take in any timestep and value (not just a sequential input), and robust scaling. Our main point is that *training* a universal forecasting model is highly nontrivial. Notably, the form and the hyperparameters of the synthetic data distribution, especially the noise parameters, must be set precisely in order for the model to achieve non-degenerate performance.
> > >
> > > Thank you very much once again, for your time in reviewing our paper.

---

> > > ### Author Response · Authors · 2023-08-19
> > > **Second reply to 3SYZ, part 3**
> > >
> > > Here, we include results with the new "Weekly" baseline (from our answer to question 3). It often achieves 2nd place behind ForecastPFN in the low train budget and low runtime settings.
> > >
> > > Train Budget vs. MSE Wins:
> > > | Model         |   50.0 |   100.0 |   150.0 |   200.0 |   250.0 |   300.0 |   500.0 |
> > > |:--------------|-------:|--------:|--------:|--------:|--------:|--------:|--------:|
> > > | Arima         |      0 |       0 |       0 |       0 |       0 |       0 |       0 |
> > > | Autoformer    |      1 |       3 |       8 |       9 |       8 |      10 |       6 |
> > > | FEDformer     |      2 |       4 |       5 |       7 |      13 |      **16** |      **16** |
> > > | ForecastPFN   |     **32** |      **30** |      **26** |      **23** |      **20** |      11 |      11 |
> > > | Informer      |      0 |       0 |       1 |       0 |       1 |       8 |      12 |
> > > | Last          |      9 |       9 |       9 |       9 |       9 |       9 |       9 |
> > > | Mean          |      5 |       4 |       3 |       1 |       2 |       0 |       0 |
> > > | Meta-N-BEATS  |      0 |       0 |       0 |       0 |       0 |       0 |       0 |
> > > | Prophet       |      0 |       0 |       0 |       0 |       0 |       0 |       0 |
> > > | Weekly |     13 |      12 |       8 |       8 |       6 |       3 |       2 |
> > > | Transformer   |      0 |       0 |       2 |       5 |       5 |       4 |       5 |
> > >
> > > Time Budget vs. MSE Wins:
> > > | Model         |   1.0 |   5.0 |   10.0 |   15.0 |   30.0 |   45.0 |   60.0 |   120.0 |
> > > |:--------------|------:|------:|-------:|-------:|-------:|-------:|-------:|--------:|
> > > | Arima         |     0 |     0 |      0 |      0 |      0 |      0 |      0 |       0 |
> > > | Autoformer    |     0 |     3 |      4 |      5 |      9 |      6 |      6 |       6 |
> > > | FEDformer     |     2 |     5 |     10 |     11 |     10 |     **18** |     **18** |      **18** |
> > > | ForecastPFN   |    **30** |    **23** |     **21** |     **19** |     **14** |     12 |     11 |      11 |
> > > | Informer      |     1 |     1 |      3 |      4 |      9 |     10 |     10 |      10 |
> > > | Last          |    10 |     9 |      9 |      9 |      9 |      9 |      9 |       9 |
> > > | Mean          |     5 |     4 |      3 |      2 |      1 |      0 |      0 |       0 |
> > > | Meta-N-BEATS  |     0 |     0 |      0 |      0 |      0 |      0 |      0 |       0 |
> > > | Prophet       |     5 |     5 |      5 |      5 |      5 |      4 |      4 |       4 |
> > > | Weekly |     9 |     7 |      3 |      2 |      1 |      0 |      0 |       0 |
> > > | Transformer   |     1 |     6 |      5 |      6 |      5 |      4 |      4 |       4 |
> > >
> > >
> > > Train Budget vs. MSE Rank:
> > > | Model         |   50.0 |   100.0 |   150.0 |   200.0 |   250.0 |   300.0 |   500.0 |
> > > |:--------------|-------:|--------:|--------:|--------:|--------:|--------:|--------:|
> > > | Arima         |   7.44 |    8.29 |    8.59 |    7.96 |    7.97 |    8.26 |    8.26 |
> > > | Autoformer    |   6.33 |    5.52 |    3.65 |    3.33 |    3.69 |    2.78 |    3.26 |
> > > | FEDformer     |   4.89 |    4.78 |    4.43 |    3.3  |    **2.65** |    **2.54** |    **2.83** |
> > > | ForecastPFN   |   **2.03** |   **2.42** |    **2.85** |    **2.98** |    3.13 |    3.47 |    3.56 |
> > > | Informer      |   7.71 |    6.21 |    6.15 |    6.26 |    6.04 |    4.9  |    4.27 |
> > > | Last          |   2.78 |    3.04 |    3.71 |    4.3  |    4.35 |    4.94 |    5.01 |
> > > | Mean          |   2.36 |    2.68 |    3.15 |    3.72 |    3.98 |    4.56 |    4.64 |
> > > | Meta-N-BEATS  |   3.2  |    3.41 |    4.14 |    4.7  |    4.72 |    5.29 |    5.41 |
> > > | Prophet       |   9.11 |    9.13 |    8.9  |    8.88 |    8.97 |    8.83 |    8.96 |
> > > | Weekly |   2.95 |    3.07 |    3.95 |    4.51 |    4.61 |    5.25 |    5.15 |
> > > | Transformer   |   6.12 |    6.33 |    5.47 |    5.05 |    4.9  |    4.19 |    3.64 |
> > >
> > > Time Budget vs. MSE Rank:
> > > | Model         |   1.0 |   5.0 |   10.0 |   15.0 |   30.0 |   45.0 |   60.0 |   120.0 |
> > > |:--------------|------:|------:|-------:|-------:|-------:|-------:|-------:|--------:|
> > > | Arima         |  5.03 |  6.78 |   7.23 |   7.53 |   8.19 |   8.34 |   8.28 |    8.38 |
> > > | Autoformer    |  6.83 |  5.72 |   5.31 |   5.22 |   3.83 |   3.29 |   3.26 |    3.32 |
> > > | FEDformer     |  4    |  4.28 |   3.6  |   3.34 |   **3.45** |   **2.85** |   **2.72** |    **2.77** |
> > > | ForecastPFN   |  **2.03** |  **2.75** |   **2.9**  |   **3.04** |   **3.45** |   3.6  |   3.59 |    3.66 |
> > > | Informer      |  5.86 |  5.99 |   5.63 |   5.72 |   4.74 |   4.48 |   4.38 |    4.33 |
> > > | Last          |  2.78 |  3.66 |   4.06 |   4.18 |   4.85 |   5.06 |   5.07 |    5.13 |
> > > | Mean          |  2.42 |  3.36 |   3.79 |   3.84 |   4.44 |   4.63 |   4.58 |    4.76 |
> > > | Meta-N-BEATS  |  3.21 |  4.13 |   4.52 |   4.6  |   5.24 |   5.46 |   5.47 |    5.53 |
> > > | Prophet       |  6.09 |  7.28 |   7.62 |   7.66 |   7.9  |   7.95 |   7.89 |    8.01 |
> > > | Weekly |  3.05 |  3.87 |   4.27 |   4.36 |   5.04 |   5.31 |   5.3  |    5.4  |
> > > | Transformer   |  6.12 |  6.3  |   5.77 |   5.4  |   3.77 |   3.98 |   4.32 |    3.71 |

---

> > > > ### Comment · Reviewer_3SYZ · 2023-08-19
> > > >
> > > > Thanks authors for the extensive reply, I have raised my score to borderline accept since my concerns about fairness of comparison in the setting of low train time has been addressed. My reason for not giving a higher score is that I am still doubtful about the low train time setting -- the out of distribution case is different from low train time setting. If OOD is our concern, then we should consider OOD methods... such as AdaRNN and many more.
> > > >
> > > > I think there is a huge onus on the authors' end to update the papers with much more extensive empirical results.  I acknowledge that FLOPs is also not sufficient alone, as such reporting results on both wall clock and FLOPs would give readers a much better understanding of the comparisons. I would also like to point out that existing Transformer architectures for forecasting can already take in any timestep and value, based on the existing formulations.

---

> > > > > ### Author Response · Authors · 2023-08-20
> > > > >
> > > > > Thank you for your reply. We are glad that we addressed your points about the comparisons in the low train time setting, and we are happy to see your more favorable view of our work.
> > > > >
> > > > > We would like to very briefly mention a couple more points regarding the low train time setting.
> > > > >
> > > > > We agree that the more compelling setting, which has more applications in the real world, is the low train size setting. We believe that ForecastPFN’s success in the low train size setting is a notable contribution by itself, and we would be happy to make the necessary edits to emphasize this setting slightly more than the low runtime setting (although we think the low runtime setting is still important to include in the paper).
> > > > >
> > > > > We agree with you that FEDFormer and the other models can be reused many times without retraining, and we also agree that models such as AdaRNN can handle the out of distribution case. Instead of “out of distribution”, a better phrase for us to use would have been “new dataset” or “different distribution.” We were thinking of applications that require many low runtime / low computation forecasts which are all very different from one another.
> > > > >
> > > > > We are currently still working on using FLOPs as a metric our experiments. For now, we can report the parameter count of the deep learning models, which is very roughly correlated with FLOPs.
> > > > >
> > > > > | Model | #Params  |
> > > > > |---| ---|
> > > > > | Autoformer | 1.05E7 |
> > > > > | FEDFormer | 1.44E7 |
> > > > > | Informer | 1.13E7 |
> > > > > | Transformer | 1.05E7 |
> > > > > |ForecastPFN | 1.50E6 |
> > > > >
> > > > > Finally, we also included a preliminary [ablation study](https://openreview.net/forum?id=tScBQRNgjk&noteId=IkeN4k7sXF) on the scale of the noise in the synthetic data.
> > > > >
> > > > > Thank you once again for your time!

---

### Official Review · Reviewer_5AR9 · 2023-07-04

**Soundness:** 2 fair
**Presentation:** 3 good
**Contribution:** 2 fair
**Rating:** 5
**Confidence:** 3

**Summary:**

This paper proposes a zero-shot prior-data fitted network (PFN) for time-series forecasting. Existing works have challenges in designing a general and flexible PFN for general time-series distributions, and tuning an architecture and training scheme. This work overcomes these by designing a novel synthetic time-series distribution in training and using a transformer architecture for queries to arbitrary future timestep. Experiments have been presented to demonstrate the improved performance of the proposed model over comparison models.

**Strengths:**

1. The proposed work tackles an interesting setting where few initial observations are available in forecasting.
2. The transformer-based network aims to predict arbitrary future time steps and the robust scaling deals with the extreme scale of time series, which increase the novelty of the methodology.
3. The results seem compelling as well, with a clear outperformance of the proposed model over comparison baselines.

**Weaknesses:**

1. In Section 3.2, it is not clear what motivates the author to choose a multiplicative data generation model with seasonal and trend components as the synthetic prior. It would be appreciated if the author could elaborate more on that. Also, the ablation study could be more sufficient: the diversity of the synthetic data generated for training should be explored since the model relies on a general synthetic data distribution.
2. The author could provide more details about the architecture in Section 3.3: 1) How does the network learn diverse temporal features of the input data? 2) How does the network deal with different input sizes? 3) What is the benefit of using transformer-based layers vs residual blocks or recurrent neural networks?
3. In Section 4, the author should explain the definition of train budget and time budget. Are highlighted texts “training time” and “training data” related to them? It would be better to make them consistent.


**Questions:**

Please see the weaknesses mentioned above.

**Limitations:**

Yes.

---

> ### Author Rebuttal · Authors · 2023-08-10
>
> Thank you for the excellent feedback. We are glad to see that you are positive towards our work, including our novel methodology and our results. We also find that your comments will substantially improve the final version of our paper. We reply to each question below.
>
> **W1.1. Multiplicative data generation model.**
> Yes, additive vs. multiplicative noise is an interesting question. We chose multiplicative noise to better balance the amount of signal to noise across all series. If we used additive noise, then the impact of the noise depends on the ratio of the scale of the base series and the scale of the noise. Since our base series have linear and exponential terms, additive noise would cause the signal-to-noise ratio to vary substantially over series with a trend component. Therefore, we used multiplicative noise, which is a simpler parametrization to achieve a signal to noise ratio for a series. We now updated our paper to include this discussion, and we agree that including an ablation study is a good idea for future work.
>
> **W1.2. Ablation on the diversity of synthetic data.**
> This is a great idea. Since each training procedure takes 30 GPU hours, it is challenging to do a thorough ablation study. As mentioned in Section 3.3, we performed light HPO on the synthetic data parameterization in the early stages of the project, using the training loss as a signal. We are currently working on a simple ablation study on the noise of the synthetic data, which we will finish during the NeurIPS discussion period.
>
> **W2.1. How does the network learn diverse temporal features.**
> This is a good question. We chose to use a transformer architecture because there is a large body of empirical evidence across different areas of research that transformers learn spatio-temporal features (e.g., vision, speech and language). We used a fairly lightweight transformer (one multi-head attention layer and two feedforward layers; details in Section 3.3), and our results show that it learns diverse temporal trends. In fact, a concrete demonstration that we can do in the future would be to train our transformer to approximate a partial Fourier transform.
>
> **W2.2 How does the network deal with different input sizes.**
> Again, since we use a transformer, it can easily deal with variable input lengths. In our experiments, we set a maximum input length to 200 just because of computation constraints that we imposed. A good experiment in future work would be to increase the maximum input length to, e.g., 5000 inputs, although this would be substantially slower.
>
> **W2.3 What is the benefit of using transformer-based layers vs residual blocks or recurrent neural networks.**
> Note that ForecastPFN is a fully pretrained model – similar to an LLM, but for forecasting. In other research areas that create “foundation models” for language modeling and even computer vision, transformers are by far the most common choice. However, as you say, we could also have tried out other types of architectures. One other reason is that transformers are currently very fast on hardware due to receiving a lot of attention from the research community. Furthermore, other models such as RNNs are inherently sequential and harder to saturate GPU compute; therefore, transformers are the most efficient and best-performing choice for ForecastPFN.
>
> **W3. Make definitions consistent.**
> Thank you. We will clarify this point in Section 4 better. We define training time as the amount of wall clock time that is permitted for each model to train (1 to 120 seconds) and training budget as the amount of data points given to each model to train (50 to 500). This is given on lines 242-244 in the present manuscript, but we have rewritten the description to be more clear.
>
> Thank you very much, once again, for your positive view of our work and your excellent suggestions. If you find our responses satisfying, since we addressed your weaknesses, we respectfully ask that you consider increasing your score. On the other hand, we would be very happy to answer any follow-up or additional questions you have.

---

> ### Author Response · Authors · 2023-08-20
> **Quick reminder and new ablation**
>
> Dear reviewer 5AR9, thank you once again for your insightful review. We appreciate that you find our work is novel, that it tackles an interesting setting, and that we achieve compelling results.
>
> We would like to check in to see if you have any follow-up comments on our rebuttal. We replied to your three weaknesses on the motivation for a multiplicative data model, on the details of the architecture, and on the definitions of train budget and time budget.
>
> We would also like to mention that we now have preliminary results for a synthetic data ablation study (which you mentioned in part 1 of your weaknesses). We trained ForecastPFN with three scales of our noise parameter,  $m_{\text{noise}}$: 1 (original ForecastPFN), $2/3$ (low noise), and $1/6$ (lowest noise). We do not change the scale of the noise in the validation data. Below, we report the train and validation MSE on synthetic data for each model.
>
> Training MSE Loss:
> | Epoch | 0 | 1 | 25 | 50 | 100 | 200 | 300 |
> |---| ---| ---| ---| ---| ---| ---| ---|
> | ForecastPFN with lowest noise | 23.71 | 5.009 | 2.326 |1.991 | 1.484 | 0.8781 | 0.5912 |
> | ForcastPFN with low noise| 0.3593 | 0.294 | 0.08648 | 0.08536 | 0.04981 | 0.03857 | 0.3835 |
> | ForecastPFN| **0.2074** | **0.1061**| **0.06425** | **0.06642** | **0.04882** | **0.03696** | **0.03196** |
>
> Validation MSE Loss:
> | Epoch | 0 | 1 | 25 | 50 | 100 | 200 | 300 |
> |---| ---| ---| ---| ---| ---| ---| ---|
> | ForecastPFN with lowest noise | 1.13E6 | 1.58E5 | 2.78E4 | 1.31E5 | 7.12E4 | 4760 | 1.35E4 |
> | ForcastPFN with low noise | 5.06E4 | 2.65E4 | **3.09E4** | **1.09E6** | 3.52E4 | 4487 | 1.57E4 |
> | ForecastPFN | **4.94E4** | **2.28E4** | 4.2E4 | 1.88E5 | **1.66E4** |  **4267** | **1.22E4** |
>
> Recall that we remove noise in the ground-truth *predictions* of the training data as a design decision that improves performance (lines 200-203 and Appendix E.6), which is why the scale looks different for the train and validation losses. This new ablation study complements our ablation study in Appendix E.6 which considers noise in the predictions of the training data. We will continue updating this ablation study by adding more values of noise and more parameters.
>
> Please let us know if you have any additional questions or comments. Thank you!

---

> > ### Comment · Reviewer_5AR9 · 2023-08-20
> >
> > Thank you for addressing my concerns and adding experiments.

---

### Official Review · Reviewer_MwwS · 2023-07-06

**Soundness:** 3 good
**Presentation:** 3 good
**Contribution:** 3 good
**Rating:** 5
**Confidence:** 4

**Summary:**

ForecastPFN is a zero-shot forecasting model designed to overcome the limitations of traditional forecasting methods when dealing with data-sparse applications. Unlike most approaches, ForecastPFN is trained solely on synthetic data, which captures diverse time series patterns and incorporates multi-scale seasonal trends, global trends, and noise. This prior-data fitted network approximates Bayesian inference and enables accurate and fast predictions in a single pass. Extensive experiments presented in the paper demonstrate that ForecastPFN outperforms state-of-the-art methods across multiple datasets and forecast horizons, even when the latter are trained with significantly more data points.

**Strengths:**

1. The manuscript is well-written, and the presented ideas are easy to follow.
2. The concept of training a foundational time series model solely on synthetic data is innovative and exciting. This approach not only addresses the issue of lower forecasting accuracy in data-sparse scenarios but also has positive environmental implications.
3. The execution of this work is commendable. The paper provides detailed information on the generation of the synthetic dataset, the model architecture, and the training process. Additionally, the authors conducted an extensive set of experiments, exploring various model configurations, prediction horizons, and datasets from multiple domains

**Weaknesses:**

The paper predominantly relies on plots to present the results, without including tables. While it is understandable that summarizing all the experiments in tabular form would be challenging due to the extensive nature of the research, it can be difficult for readers to extract meaningful information from a large number of plots. Additionally, certain plots, such as Fig 5, are scaled down to a point where they become challenging to interpret. The same issue extends to the plots presented in the appendix. To enhance clarity, it is recommended that the authors include some key results in tabular form and adjust the scale of the plots to improve readability for readers.

**Questions:**

1. Considering the focus on commercial forecasting applications in this work, it would be valuable to understand the reasons behind modeling ForecastPFN as a point forecaster instead of a probabilistic forecaster.
2. In real-world forecasting applications, the presence of exogenous features can provide additional information that directly affects the accuracy of forecasts. How does ForecastPFN handle the incorporation of exogenous features?
3. It would be interesting to know if the authors explored the option of fine-tuning ForecastPFN on the target data. This analysis could provide insights into the impact of fine-tuning on the performance of ForecastPFN in downstream forecasting tasks.

**Limitations:**

See above

---

> ### Author Rebuttal · Authors · 2023-08-10
>
> Thank you for the excellent feedback. We are glad to see that you find our idea innovative and exciting, and that you find the execution commendable. We also find that your comments will substantially improve the final version of our paper. We reply to each question below.
>
> **W1: No tables.** Thank you, we can easily add tables for all of our main experiments, and we are happy to do so. We present tables below which show the superiority of ForecastPFN on a majority of datasets at different train budgets with a prediction length of 24.
> We have printed tables with MSE values for Train Budget = 50 below.
>
> |     |         ECL |        ETT1 |        ETT2 |    exchange |         ili |     traffic |     weather |
> |-------------|------------:|------------:|------------:|------------:|------------:|------------:|------------:|
> | Arima       |       1.844 |       0.344 |       1.590 |       1.175 |       5.039 |       2.696 |       0.041 |
> | Autoformer  |       2.368 |       0.850 |       1.299 |       0.381 |       1.858 |       3.803 |       0.371 |
> | FEDformer   |       0.899 |       0.762 |       1.175 |       0.786 |       1.549 |       2.331 |       0.817 |
> | ForecastPFN |       1.104 | **0.121** | **0.340** |       0.061 | **1.102** |       2.029 | **0.010** |
> | Informer    |       1.078 |       1.967 |       1.105 |       4.694 |      11.045 |       5.848 |      0.325 |
> | Last        |       0.889 |       0.191 |       0.492 | **0.024** |       1.487 |       3.030 |       0.017 |
> | Mean        | **0.658** |       0.174 |       0.649 |       0.042 | **1.102** | **1.558** |       0.012 |
> | Metalearn   |       0.871 |       0.192 |       0.500 |       0.025 |       1.572 |       2.795 |       0.014 |
> | Prophet     |       2.181 |       3.298 |      14.060 |     421.577 |      11.813 |       2.328 |       0.101 |
> | Transformer |       0.827 |       0.630 |       0.538 |       1.833 |       5.578 |       3.093 |       0.182 |
>
>
> And with Train Budget = 500 below.
> |     |         ECL |        ETT1 |        ETT2 |    exchange |         ili |     traffic |     weather |
> |-------------|------------:|------------:|------------:|------------:|------------:|------------:|------------:|
> | Arima       |       1.974 |       0.201 |       1.075 |       1.175 |       4.852 |       1.628 |       0.041 |
> | Autoformer  | **0.482** |       0.141 |       0.360 |       0.073 | **0.705** | **0.473** |       0.189 |
> | FEDformer   |       0.499 |       0.135 |       0.374 |       0.072 |       0.727 |       0.455 |       0.208 |
> | ForecastPFN |       1.104 | **0.121** | **0.340** |       0.061 |       1.102 |       2.029 | **0.010** |
> | Informer    |       0.442 |       0.144 |       0.263 |       0.518 |       4.978 |       1.028 |       0.186 |
> | Last        |       0.889 |       0.191 |       0.492 | **0.024** |       1.487 |       3.030 |       0.017 |
> | Mean        |       0.658 |       0.174 |       0.649 |       0.042 |       1.102 |       1.558 |       0.012 |
> | Metalearn   |       0.871 |       0.192 |       0.500 |       0.025 |       1.572 |       2.795 |       0.014 |
> | Prophet     |      15.698 |       3.226 |       4.907 |      13.347 |       3.478 |       1.530 |       0.056 |
> | Transformer |       0.621 |       0.157 |       0.277 |       0.351 |       3.576 |       0.804 |       0.012 |
>
>
> All other tables have been put in the appendix of the paper.
>
> **Q1. Probabilistic forecaster.**
> This is a great suggestion. Since this work already introduces a new paradigm, we wanted to start with the simplest regression setting, but we agree that adding probabilistic predictions is an exciting and natural area for future work. Towards that direction, we are training three ForecastPFN models to show that we can achieve probabilistic forecasts by ensembling (which is a strong technique for uncertainty calibration). For future work, it will be interesting to have the model itself output probabilities.
>
> **Q2. Exogenic features.** This is another great suggestion. Once again, we started out with a simpler setting so that we could clearly demonstrate our approach, and so that we could compare to the main experiments in prominent existing work such as FEDFormer and Informer. We think this is a great idea for future work.
>
> **Q3. Fine-tuning.** While this is once again a nice suggestion, we make three points. First, one of our main contributions is to create a universal forecasting method that can perform well on any new series, without any fine-tuning. Second, we believe that the settings where ForecastPFN stands out the most compared to prior work, is when there is very little inference time available and/or when there is very little training data available -- settings in which it would be very challenging to fine-tune. Third, we note that prior-data fitted networks (PFNs) are still a new research area just one year old, and fine-tuning has not yet been tried even for classification-based PFNs. From a theoretical standpoint, it is not clear that fine-tuning would maintain the desirable Bayesian properties of PFNs. Although, intuitively we agree that fine-tuning could help in practice.
>
> Thank you very much, once again, for your positive view of our work and your excellent suggestions. If you find our responses satisfying, since we addressed your weaknesses, we respectfully ask that you consider increasing your score. On the other hand, we would be very happy to answer any follow-up or additional questions you have.

---

> > ### Comment · Reviewer_MwwS · 2023-08-19
> >
> > Thanks for the rebuttal and doing the additional experiments. I would suggest moving these tables into the main paper as well. After reading the other reviews and the corresponding rebuttal discussions, I would like to maintain my original score.

---

> > > ### Author Response · Authors · 2023-08-20
> > > **Second reply to MwwS**
> > >
> > > Thank you very much once again for your review. We appreciate that you found our work innovative and exciting, with positive environmental implications, and also that our work is well-written and easy to follow with extensive experiments.
> > >
> > > Thank you for your update. Yes, since you listed the lack of tables as a primary weakness, we created tables with [raw MSE values](https://openreview.net/forum?id=tScBQRNgjk&noteId=Z6mb79DbuS), [MSE wins and MSE rank](https://openreview.net/forum?id=tScBQRNgjk&noteId=WZZvHCWdd0), and [percentage completion](https://openreview.net/forum?id=tScBQRNgjk&noteId=F2RWvstySO), and we added all of these into our paper.
> > >
> > > We would also like to mention that we just added a preliminary [ablation study](https://openreview.net/forum?id=tScBQRNgjk&noteId=IkeN4k7sXF) on the noise of our synthetic data, and we included [parameter counts](https://openreview.net/forum?id=tScBQRNgjk&noteId=OexP4WpgRv) for the deep learning models.
> > >
> > > Please let us know if you have any additional follow-up questions or reservations about our work, and we will be happy to reply. Thank you for your time!

---

### Author Rebuttal · Authors · 2023-08-10

We thank all reviewers for their valuable feedback and suggestions. Our work introduces the first synthetically-trained, zero-shot, universal forecasting model, which performs particularly well in low-resource settings (small amount of in-distribution data, and/or low inference time budget), which are very important settings in practice. We appreciate that the reviewers find our method novel and exciting (​​**MwwS, 5AR9, 3SYZ**), our paper well-written and easy to follow (**MwwS, 3SYZ, 458V**), our experiments extensive and compelling (**5AR9, 3SYZ**), and the low-resource setting interesting and impactful (**MwwS, 5AR9, 458V**).

We would like to go over a few small points of confusion, just so that we are all on the same page. At a high level, we found that since our work is a substantially different paradigm from most prior work (which most reviewers mentioned as an exciting direction), there was also some confusion about our method and experimental setup. Please see our points below.
- **ForecastPFN was trained only once, ever.** Similar to foundation models such as GPT4, we only needed to train ForecastPFN once, and the weights are never updated even when evaluating on new, unseen time series.
- **ForecastPFN is a _universal_ forecasting model**. ForecastPFN, a single model, can be used for different input lengths, prediction lengths, and frequencies. It empirically performs well in different settings such as currency exchange, weather, consumer electricity usage, and illness trends. The time to output predictions for a brand new series is under one second -- just a model evaluation. To the best of our knowledge, ForecastPFN is the first model to have this level of universality, and the first to be pretrained on synthetic data.
- **We overcame significant technical challenges to create a universal, pretrained model**, such as handling dynamic scaling, designing a general and flexible architecture, and carefully designing and parameterizing the synthetic data, especially the noise distribution. **Other architectures such as FEDFormer are not able to pretrain with this level of generality** (see below).

We would also like to highlight the additional experiments we finished during the rebuttal period.
- **The FEDFormer architecture (the second-best algorithm in our experiments) cannot be pretrained with our synthetic data.** Based on the suggestions of reviewer 3SYZ, we (attempted to) pretrain FedFormer using our synthetic data. Although we spent significant time during the week-long rebuttal period, we so far are not able to get the FEDFormer architecture to achieve non-degenerate performance on the real-world datasets. Note that as we mentioned in a point above, it is non-trivial to design and train a model to achieve zero-shot, universal forecasting performance, and the ForecastPFN architecture was specifically designed for this goal.
- **We included several new tables of raw MSE values, as well as table versions of our main figures**. Based on the suggestions of reviewers MwwS and 3SYZ, we included raw MSE values as well as more tables, to better view our results. See these tables in [our reply to Reviewer MwwS](https://openreview.net/forum?id=tScBQRNgjk&noteId=Z6mb79DbuS) and in our one-page pdf.
- **We added two very simple baselines: historical mean, and previous value.** See the figures in our one-page pdf. We see that the historical mean baseline does very well, achieving second-best performance behind ForecastPFN when there is very low train budget or train time. This is a testament to how challenging it is to predict with very little training data or very little training time.

We would be very happy to keep the discussion going for any points that might be still unclear or any new comments. Thank you very much, once again, for giving suggestions for our paper.

---

> ### Author Response · Authors · 2023-08-21
> **Author update**
>
> We thank all four reviewers for replying during the discussion period.
>
> We were happy to complete all suggestions, such as adding 3 new baselines to our experiments, adding new ablation studies, adding tables of the raw MSE values of our experiments, and answering clarification questions about our experimental setup and synthetic data.
>
> We once again appreciate that all reviewers found our work novel and exciting: the first forecasting model to be trained purely on synthetic data. Thank you once again, for all of your comments and suggestions!

---

> > ### Comment · Area_Chair_necU · 2023-08-22
> >
> > Thank you for making an effort to address the reviewers' concerns which includes conducting more experiments.
> >
> > AC

---

### Decision · Program_Chairs · 2023-09-21

**Decision:**

Accept (poster)

**Comment:**

This paper proposes a new approach for zero-shot forecasting in data-sparse applications by training a model purely on a novel synthetic data distribution with prior distribution similar in properties to real-world data.

It is clear that, if successful, this approach will be very useful for data-sparse applications. The idea is also interesting and has not received much attention in the past. Extensive experiments are reported showing significant improvement over the baselines. Moreover, the paper is generally well written. However, it does have much room for improvement as reflected by the comments and questions raised by the reviewers. We thank the authors for trying to answer all the questions raised and conducting further experiments when answering some of the questions.

While we generally agree that this work has merits to be accepted, we hope the authors will take the comments and suggestions seriously to revise the paper to make it appeal better to the research community.